# OmniField: Conditioned Neural Fields for Robust Multimodal Spatiotemporal Learning

**Kevin Valencia[1], Thilina Balasooriya[2], Xihaier Luo[3], Shinjae Yoo[3] & David Keetae Park[3]**
[1]UCLA    [2]Columbia University    [3]Brookhaven National Laboratory
`kevinval04@ucla.edu, tnb2119@columbia.edu,`
`{xluo,sjyoo,dpark1}@bnl.gov`

## Abstract

Multimodal spatiotemporal learning on real-world experimental data is constrained by two challenges: within-modality measurements are sparse, irregular, and noisy (QA/QC artifacts) but cross-modally correlated; the set of available modalities varies across space and time, shrinking the usable record unless models can adapt to arbitrary subsets at train and test time. We propose **OmniField**, a continuity-aware framework that learns a continuous neural field conditioned on available modalities and iteratively fuses cross-modal context. A multimodal crosstalk block architecture paired with iterative cross-modal refinement aligns signals prior to the decoder, enabling unified reconstruction, interpolation, forecasting, and cross-modal prediction without gridding or surrogate preprocessing. Extensive evaluations show that OmniField consistently outperforms eight strong multimodal spatiotemporal baselines. Under heavy simulated sensor noise, performance remains close to clean-input levels, highlighting robustness to corrupted measurements.

## 1 Introduction

Spatiotemporal data gathered from scientific experiments and observations are inherently multimodal. In climate science, for instance, measurements of temperature, humidity, and wind speed are collected concurrently to model atmospheric dynamics (Hersbach et al., 2020). Similarly, air pollution studies rely on data from various sensors measuring particulate matter, ozone, and nitrogen oxides (U.S. Environmental Protection Agency, 2025). This multimodality extends across numerous fields, including materials science, where stress and strain are measured together (Sutton et al., 2009), particle physics, which analyzes energy deposits and particle tracks simultaneously (ATLAS Collaboration, 2008), and biology, where multimodal imaging captures diverse cellular processes (Bischof et al., 2024). Yet two practical challenges persist (Fig. 1):

- **Data challenge:** measurements within each modality are sparse and irregular, with QA/QC noise that can be sensor- or instance-specific (Cressie & Wikle, 2011).
- **Modality challenge:** coverage and fidelity vary across modalities. Sensors sit at different locations, exhibit distinct sparsity patterns, and have modality-specific noise structures that can shift over space, time, and even individual devices (Hall & Llinas, 1997).

Prior efforts regarding these two challenges can be reviewed from two complementary perspectives: *data* and *model*. From the *data* perspective, pre-processing techniques like filtering, gridding/kriging, and imputation regularize irregular, noisy samples by constructing a surrogate dataset before learning. While this can stabilize downstream training, it introduces systematic side effects like smoothing bias, which attenuates extremes and high-frequency structure, and uncertainty collapse, as the surrogate is treated as ground truth and the uncertainty from these guesses is not carried forward in the analysis (Little & Rubin, 2002). From the *model* perspective, methods designed for irregular sampling like continuous-time latent models, neural ODEs, and graph-dynamic formulations respect non-gridded timing and can reduce reliance on heavy pre-processing. However, they typically assume fixed observation operators and a shared sampling index across modalities. In practice, sensor supports, coverage, and noise characteristics vary by modality, location, and time,

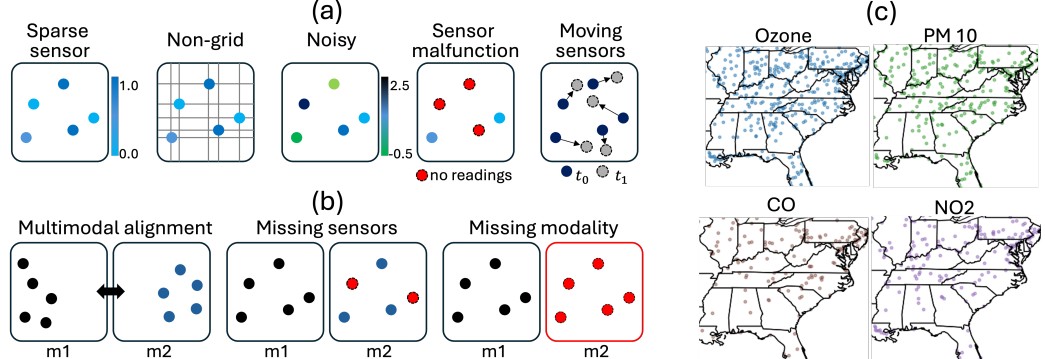

Figure 1: (a) Data challenges: sparse, irregular, noisy, and dynamic measurements. (b) Modality challenges: misaligned supports, modality-specific noise, and variable modality availability. (c) Real-world example: ambient air pollution data collected from hundreds of monitors.

yielding likelihood misspecification when these assumptions are violated (Chen et al., 2023; Gravina et al., 2024; Feng et al., 2024).

We address these challenges with **OmniField**, a unified continuity-aware framework that extends the principles of conditioned neural fields (CNFs) (Xie et al., 2022). Our approach first builds a robust, high-fidelity CNF backbone capable of handling the *data challenges* and then introduces a novel, iterative fusion architecture and training strategy to address the *modality challenges*. Our contributions are as follows:

- We introduce **OmniField**, a continuity-aware multimodal conditioned neural field for complex scientific observational data without gridding or surrogate preprocessing.
- We demonstrate two novel mechanisms for learning spatiotemporal multimodal correspondences while robust to noise: a multimodal crosstalk (MCT) block for cross-modal information exchange and iterative cross-modal refinement (ICMR) that progressively aligns signals.
- We carry out extensive experiments which demonstrates the proposed model outperforms eight strong baselines, yielding a 22.4% average relative error reduction across our benchmarks, and remains near-clean under heavy simulated sensor noise, confirming robustness to corrupted measurements
- We contribute ClimSim-LHW to reflect realistic observational sparsity and an ML-ready EPA-AQS dataset, enabling systematic evaluation under data- and modality-challenge regimes.

## 2    RELATED WORKS

**Neural Fields.** Neural fields, implicit, coordinate-based networks, encode continuous functions that map coordinates to signals, enabling compact, high-frequency detail and resolution-free sampling (Xie et al., 2022; Sitzmann et al., 2020). They underpin high-fidelity reconstruction and synthesis across vision and science, with canonical instances including NeRF for novel view synthesis and signed/occupancy fields for geometry (Mildenhall et al., 2020; Park et al., 2019).

**Conditioned Neural Fields.** Conditioned neural fields (CNFs) learn maps from auxiliary context (e.g., parameters, coefficients, boundary/sensor data) to continuous fields, so a single model spans a family of signals. This operator-learning view is exemplified by Fourier Neural Operators (FNOs), which learn parametric PDE solution operators efficiently in spectral space (Li et al., 2020). Transformer operator learners extend this perspective with attention over discretizations (Li et al., 2023), while neural-field formulations on general geometries demonstrate CNFs that natively accommodate complex domains (Serrano et al., 2023). Building on these trends, SCENT shows a scalable, explicitly conditioned spatiotemporal CNF for scientific data that unifies interpolation, reconstruction, and forecasting, highlighting CNFs as a practical vehicle for continuous spatiotemporal modeling (Park et al., 2025).

**Multimodal Conditioned Neural Fields.** Multimodal CNFs fuse heterogeneous evidence as conditioning signals to improve reconstruction and forecasting under sparse or partial observations. PROSE-FD illustrates this direction for physics: a multimodal PDE foundation model that learns multiple fluid-dynamics operators and leverages diverse inputs (e.g., coefficients, coarse fields, sen-

sor streams) to generalize across tasks (Liu et al., 2024). In vision, MIA meta-learns implicit neural representations with multimodal iterative adaptation, offering strong performance with limited observations but relying on bi-level optimization (Lee et al., 2024).

## 3 BACKGROUND AND SETUP

**Data.** Let $\mathcal{X} \subset \mathbb{R}^d$ be the spatial domain and $\mathcal{T} \subset \mathbb{R}$ time. We consider a fixed catalog of modalities $\mathcal{M}_{\text{all}} = \{1, 2, \ldots, M\}$. For modality $m$, a measurement at $(\mathbf{x}, t) \in \mathcal{X} \times \mathcal{T}$ is $y_m(\mathbf{x}, t)$. *Example:* in air quality, $\mathcal{M}_{\text{all}} = \{\text{PM}_{2.5}, \text{O}_3, \text{NO}_2\}$.

- *Context (inputs).* At input time $t_{\text{in}}$, each available modality $m \in \mathcal{M}_{\text{in}} \subseteq \mathcal{M}_{\text{all}}$ contributes a finite, irregular set $U_m = \{(\mathbf{x}, y_m(\mathbf{x}, t_{\text{in}}))\}$, and $C = \{U_m : m \in \mathcal{M}_{\text{in}}\}$. *Example:* if only PM$_{2.5}$ and NO$_2$ arrived, then $\mathcal{M}_{\text{in}} = \{\text{PM}_{2.5}, \text{NO}_2\}$ and $C = \{U_{\text{PM}_{2.5}}, U_{\text{NO}_2}\}$.
- *Queries (targets).* Predictions are requested at time $t_{\text{out}} = t_{\text{in}} + \Delta t$ with $0 \leq \Delta t < t_h$ (a time horizon hyperparameter), for target modalities $\mathcal{M}_{\text{out}} \subseteq \mathcal{M}_{\text{all}}$ and query locations $Q_m = \{\mathbf{x}\}$, giving $Q = \{(m, \mathbf{x}) : m \in \mathcal{M}_{\text{out}}, \mathbf{x} \in Q_m\}$. *Example:* for cross-modal prediction of ozone at current time, set $\Delta t = 0$, $\mathcal{M}_{\text{out}} = \{\text{O}_3\}$, and choose $Q_{\text{O}_3}$ (locations of interest).

**Goal.** Our objective is to build a neural network $\mathcal{F}_\theta$ that maps irregular, noisy, multimodal observations to a continuous spatiotemporal field, avoiding gridding and heavy imputation.

**Tasks.** Under this setup, the network $\mathcal{F}_\theta(\cdot)$ is designed to handle four tasks:

1. *Reconstruction:* $\Delta t = 0$, $\mathcal{M}_{\text{out}} \subseteq \mathcal{M}_{\text{in}}$, and $Q_m = \{\mathbf{x} \in U_m\}$ (predict provided observations).
2. *Spatial interpolation:* $\Delta t = 0$, $\mathcal{M}_{\text{out}} \subseteq \mathcal{M}_{\text{in}}$, and $Q_m$ contains unseen locations not in $U_m$.
3. *Forecasting:* $\Delta t > 0$ with arbitrary $Q_m$ (predict at future times).
4. *Cross-modal prediction:* some $m \in \mathcal{M}_{\text{out}} \setminus \mathcal{M}_{\text{in}}$ (predict modalities not present at input).

**Solution Strategy.** The setup above calls for a function that can be queried at arbitrary space–time coordinates. A *neural field* provides this abstraction by mapping $(\mathbf{x}, t)$ to a value via a coordinate-based network $\hat{\mathbf{y}} = \mathcal{F}_\theta(\mathbf{x}, t)$ (Xie et al., 2022). However, a vanilla neural field typically models a *single* signal (e.g., one scene or experiment) with fixed parameters, and does not directly generalize across multiple signals/instances without retraining. To model a *family* of signals with shared parameters, we adopt a *conditioned* neural field: the network takes the coordinates *and* an instance-specific summary derived from observations of that instance $\hat{\mathbf{y}} = \mathcal{F}_\theta(\mathbf{x}, t \,; c)$, where $c$ is the latent embedding encoding the identity or properties of specific signal from a set of signals.

## 4 METHOD

**Overview.** Following the solution strategy in Section 3, we introduce **OmniField**, an encoder-processor-decoder architecture (Jaegle et al., 2022; Park et al., 2025). The full model is

$$\mathcal{F}_\theta = \{\mathcal{D}_{\omega,m}\}_{m \in \mathcal{M}_{\text{all}}} \circ \mathcal{P}_\psi \circ \mathcal{E}_\phi, \qquad \theta = (\phi, \psi, \omega),$$

which maps a query $(\mathbf{x}, t)$ and a context set $C$ to modality-specific predictions. Specifically,

- **Encoder** $\mathcal{E}_\phi$ (context to local summary). Given $C$ and a query $(\mathbf{x}, t)$, the encoder builds a query-local, permutation-invariant summary $\mathbf{c}(\mathbf{x}, t) = \mathcal{E}_\phi(C \,; \mathbf{x}, t)$, aggregating irregular observations (across space, time, and modalities) into a fixed-length representation.
- **Processor** $\mathcal{P}_\psi$ (coordinates + context to latent field). The processor fuses multi-resolution coordinate encodings with the context summary: $\mathbf{h}(\mathbf{x}, t) = \mathcal{P}_\psi(\gamma(\mathbf{x}), \eta(t), \mathbf{c}(\mathbf{x}, t))$, where $\gamma, \eta$ encode spatial and temporal coordinates. This stage constitutes the conditioned neural field.
- **Decoder** $\mathcal{D}_{\omega,m}$ (latent to modality output). Each modality uses a lightweight decoder to produce predictions: $\hat{y}_m(\mathbf{x}, t) = \mathcal{D}_{\omega,m}(\mathbf{h}(\mathbf{x}, t))$.

The encoder–processor–decoder backbone provides the right scaffold, but sparse, noisy, *multimodal* observations raise three practical questions: **Q1**: how do we mitigate low-frequency bias and preserve high-frequency detail? **Q2**: how do we align signals collected on mismatched supports, scales, and noise profiles? **Q3**: how do we operate when the set of available modalities changes across space and time? Next, we address each of these questions in turn.

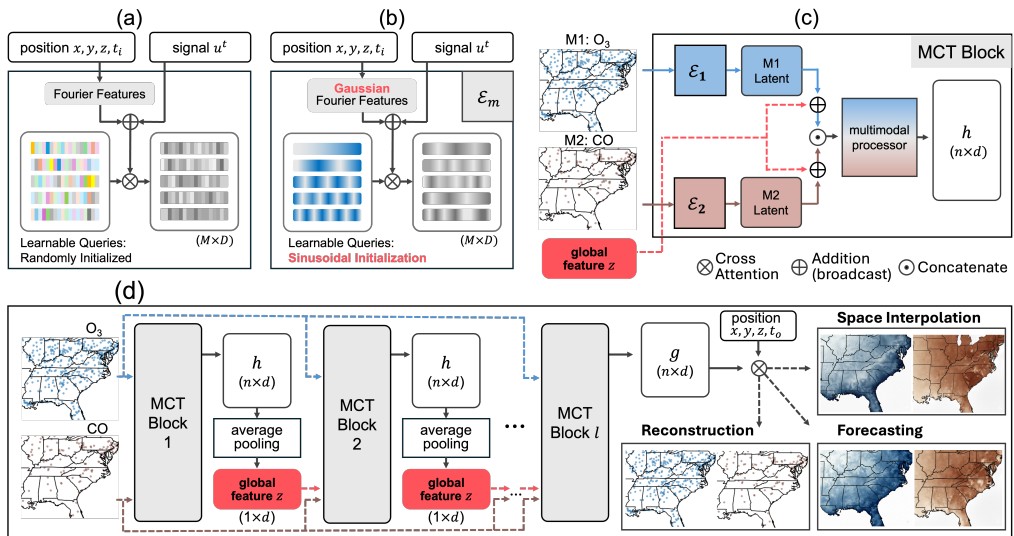

Figure 2: **Overview of Our Approach**. We illustrate (a) prior SCENT (Park et al., 2025) encoder, (b) our proposed encoder (***E***), (c) our *multimodal crosstalk* (MCT) block, and (d) our proposed OmniField architecture equipped with *iterative cross-modal refinement* (ICMR) strategy.

## 4.1 GAUSSIAN EMBEDDINGS & LEARNABLE QUERIES

Learning fine-scale structure is notoriously hard for conditioned neural fields (CNFs): training amounts to fitting a continuous function to irregular, noisy signals, and standard CNFs exhibit a low-frequency (spectral) bias. In our setting, sparse, noisy scientific observations, this bias is amplified by two factors: (i) coarse, fixed-frequency positional encodings (e.g., integer or log-stepped bands) that under-represent high frequencies; and (ii) randomly initialized learnable query tokens that provide poor initial coverage of the spectrum, creating a bottleneck for continuity-aware representations. To address this, we replace fixed sinusoidal Fourier features with **Gaussian Fourier features (GFF)**, sampling $B \in \mathbb{R}^{d \times 1}$ with entries $B_{ij} \sim \mathcal{N}(0, \sigma^2)$ so that, for a coordinate $x$, $\gamma(x) = \text{concat}\big(\cos(2\pi Bx), \sin(2\pi Bx)\big) \in \mathbb{R}^{2d}$, yielding a richer, less-biased spectral representation that better captures high-frequency detail (Tancik et al., 2020). Additionally, we introduce **sinusoidal initialization** to stabilize training and ensure balanced frequency coverage. Concretely, we initialize the $M$ learnable queries with a compact multi-scale sinusoidal pattern using log-spaced bands and unit-norm scaling ($s = d^{-1/2}$). Our experiments show that GFF and sinusoidal initialization together improves CIFAR-10 reconstruction loss and climate forecasting performance by $\times 2.74$ and $30\%$ (Table 3), respectively. Full derivations, ablations, and discussion are deferred to Appendix B.

## 4.2 MULTIMODAL ALIGNMENT

While general conditioned neural fields $\mathcal{F}(\cdot)$ provide continuity-aware representations, a single pass of encode→process→decode can under-express *fine-grained, cross-modal* correspondences when modalities differ in support, scale, and noise (Serrano et al., 2023; Park et al., 2025). Prior work such as MIA meta-learns per-instance implicit representations for sparse natural images (Lee et al., 2024); in contrast, we seek a *single-step, shared-parameter* training procedure that remains stable and efficient for spatiotemporal, multimodal data. Specifically, we introduce two advances: a *multimodal crosstalk* (MCT) module that concatenates per-modality CNF tokens and conditions the processor with a lightweight global multimodal code, and *iterative cross-modal refinement* (ICMR) that revisits this fusion over several processor steps. In effect, the model merges multimodal CNFs while being informed by global structure at each step, enabling adaptive alignment across scales and improved robustness to heterogeneous noise.

**Multimodal Crosstalk (MCT).** Let our unimodal encoder denoted as $\mathcal{E}_m$ (Fig. 2a). Our MCT outputs intermediate features $h$ as follows:

$$h := \text{MCT}\big(\{U_m^{t_{\text{in}}}\}_{m \in \mathcal{M}_{\text{in}}}, z\big) = \mathcal{P}\left(\overset{M}{\underset{m=1}{\odot}} \big[\mathcal{E}_m\big(U_m^{t_{\text{in}}}\big) \oplus z\big]\right),$$

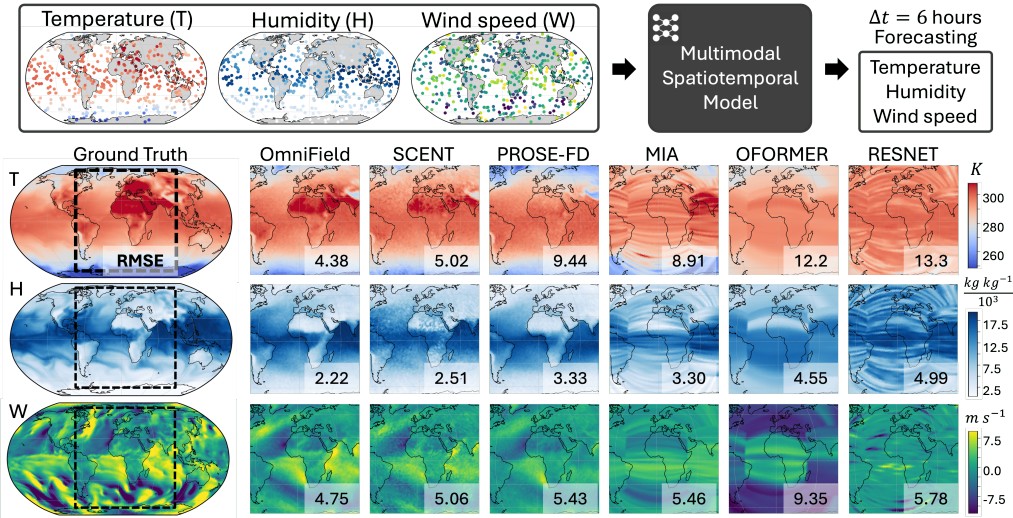

Figure 3: Qualitative Comparisons on ClimSim. Provided with a highly sparse yet multimodal observations, models generate full-field forecasting at $\Delta t = 6$ hours. We provide comparisons against the ground truth. RMSE against the Ground Truth is shown in white boxes.

where $\mathcal{E}_m(U_m^{t_{in}}) \in \mathbb{R}^{n \times d}$ denotes unimodal features, $\odot$ denotes concatenation across modalities $m$, $\oplus$ denotes addition with broadcasting, and $z \in \mathbb{R}^{1 \times d}$ represents a global feature summarizing an intermediate multimodal features. $\mathcal{P}$ denotes a multimodal processor equipped with multiple self-attention layers. The global feature $z$ serves a critical dual function; it provides global information aggregated from all multimodal inputs to facilitate cross-modal communication, while also acting as a compact information bottleneck that evolves throughout the network layers. $z$ is given from a prior layer as discussed further below.

**Iterative Cross-Modal Refinement (ICMR).** In our ICMR strategy, the global feature z acts as a communication bridge, relaying global multimodal information between the unimodal encoders. Given $\ell$ MCT blocks, we define ICMR as follows. For $k = 0, \ldots, \ell - 1$,

$$h^{(k)} := \text{MCT}\big(\{U_m^{t_{in}}\}_{m \in \mathcal{M}}, z^{(k)}\big) \in \mathbb{R}^{n \times d} \quad \Rightarrow \quad z^{(k+1)} = \frac{1}{n} \sum_{i=1}^{n} h_{i,:}^{(k)}, \tag{1}$$

Finally, the multimodal neural field is $g = h^{(\ell-1)}$. The initial global feature $z^{(0)}$ is filled with zero.

### 4.3 FLEXIMODAL FUSION

Fleximodal fusion treats modalities as $m \in \mathcal{M}$ with a presence mask $\pi_m$, enabling one model to operate on any input subset: absent channels are zero-gated at the encoders, masked out in cross-attention, and excluded from the loss (only supervised targets contribute), preventing leakage from missing inputs; this departs from training-only dropout (ModDrop; (Neverova et al., 2015)) and aligns with recent fleximodal/missing-modality methods (Han et al., 2024; Wu et al., 2024). Scientific datasets frequently have sensors or variables intermittently unavailable, making such masking essential for robust deployment. In EPA-AQS air-quality data, true daily absences naturally set $\pi_m = 0$ for some $m$, so we evaluate with native day-specific masks without imputation. For fairness, we apply the same fleximodal masking across all baselines—including *OmniField*—at train and test. We defer an in-depth discussion to Appendix C.

## 5 EXPERIMENTS

We hypothesize that a continuity-informed multimodal model, instantiated as CNFs, can address the data and modality challenges outlined in Section 1 and Fig. 1. We begin by surveying prevalent strategies for multimodal training with sparse and varying-location sensors and summarizing their comparative performance (Section 5.2). We then present our primary experiment: forecasting output

fields from sparse inputs and benchmarking against eight widely used baselines on a simulated variant of ClimSim Yu et al. (2023), a climate reanalysis dataset. Next, we validate on real data from EPA-AQS—an air-pollution network that exemplifies these challenges—again comparing the eight baselines with our approach, OmniField. Finally, ablation studies on CIFAR-10 (spatial), RainNet Ayzel et al. (2020)(spatiotemporal), and ClimSim (multimodal spatiotemporal) quantify the contributions of our modeling choices, demonstrating the efficacy and robustness of continuity-informed neural-field conditioning across data regimes.

## 5.1 Experimental Setup

**Datasets.** We evaluate our proposed method against baseline methods using four diverse datasets. Detailed information on data statistics for these datasets can be found in Appendix A.

- **ClimSim-THW**. We use a subset of 10,000 consecutive 1-hour intervals from ClimSim and retain three modalities: temperature ($T$; K), humidity ($H$; $kg\,kg^{-1}$), and wind speed ($W$; $m\,s^{-1}$). The reanalysis fields are defined over 21,600 unique sensor locations; we select the lowest of the 60 vertical levels. To study sparse, partially co-located sensing, each modality is observed on 432 locations, with a triple overlap $|T \cap H \cap W| = 108$, pairwise-only regions $|T \cap H| = |T \cap W| = |H \cap W| = 81$, and modality-exclusive regions $|T| = |H| = |W| = 162$, yielding a union of 837 unique input locations. We use the time horizon $t_h = 6$ hours.
- **EPA-AQS**. The U.S. Environmental Protection Agency's Air Quality System (EPA-AQS) is the national repository of quality-assured ambient air-pollution measurements. We use records from 1987–2017 and ingest the data as-is across six modalities—Ozone ($O_3$, ppm), $PM_{2.5}$ ($\mu/m^3$), $PM_{10}$ ($\mu/m^3$), $NO_2$ (ppm), CO ($\mu/m^3$), and $SO_2$ ($\mu/m^3$)—and parse it using calendar days as the reference timestep. Model inputs use native stations on the anchor day, and we use the time horizon $t_h = 5$ days.
- **CIFAR-10**. We use CIFAR-10 as a purely spatial benchmark. We use all 32×32-resolution 60,000 natural images for ablating the reconstruction qualities.
- **RainNet**. This dataset consists of 5-minute interval precipitation records from DWD radar composites (900×900 km domain). Following Park et al. (2025), we downsample this to 64×64 grid, and use the fixed split of 173,345 and 43,456 instances for training and validation, respectively.

**Baselines.** We benchmark OmniField against eight well-acknowledged and advanced models: UNet (Ronneberger et al., 2015), ResNet (He et al., 2016), FNO Li et al. (2020), OFormer Li et al. (2023), CORAL Serrano et al. (2023), PROSE-FD Liu et al. (2024), MIA Lee et al. (2024), SCENT (Park et al., 2025). Table 1 compares their reported capacities against our OmniField. Most baselines are inherently multimodal, hence we extend the prior architectures following the Mid-Fusion (Section 5.2, Nagrani et al. (2021)) scheme.

Table 1: Comparing model capacities in spatiotemporal learning.

| Model | Mesh Agnostic Learning | Space Continuity | Time Continuity | Single-Step Training |
|---|---|---|---|---|
| Unet | ✗ | ✗ | ✗ | ✓ |
| ResNet | ✗ | ✗ | ✗ | ✓ |
| FNO | ✗ | ✓ | ✗ | ✓ |
| OFormer | ✓ | ✗ | ✗ | ✓ |
| CORAL | ✓ | ✓ | ✓ | ✗ |
| PROSE-FD | ✗ | ✓ | ✓ | ✓ |
| MIA | ✗ | ✓ | ✗ | ✗ |
| SCENT | ✓ | ✓ | ✓ | ✓ |
| OmniField | ✓ | ✓ | ✓ | ✓ |

For models not mesh-agnostic, we take the data as a 1-dimensional vector attached with coordinate positions. All experiments were performed on a single NVIDIA H100 80GB HBM3 GPU.

## 5.2 Multimodal Gains for Sparse Scientific Data

We first attempt to answer whether multimodality enhances performance across distinctive training strategies for multimodal sparse data. We evaluate four strategies: (i) *Co-Location*—restrict each modality to the common sensors so all modalities observe the same sensors; (ii) *Interpolation*—impute missing entries so every modality is defined on the union of all sensor locations; (iii) *Mid-Fusion*—preserve the original sparsity, encode each modality separately, then fuse features mid-network to condition a single neural field; and (iv) our *ICMR*, which preserves sparsity while sharing cross-modal context before conditioning a unified neural field. We train OmniField either on a single modality (T) or three modalities (T,H,W), in both cases forecasting all three modalities.
• **Results:** Fig. 4(a) shows the results. Multimodal inputs always improves the forecasting performances across all multimodal strategies and modalities. While *Co-Location* performs worst due

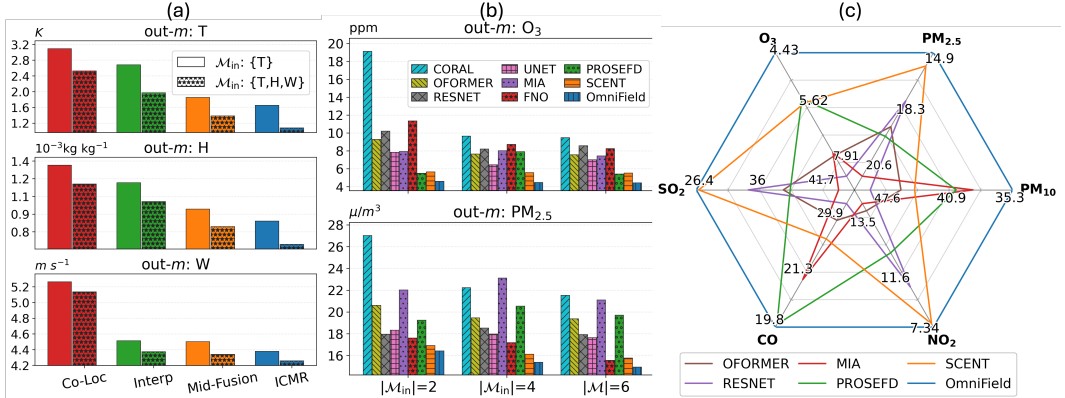

Figure 4: (a) Multimodal training results on OmniField, comparing four training strategies (Co-Loc=Co-Location; Interp=Interpolation; out-$m$=queried-out modality) on ClimSim. (b) EPA-AQS baselines are trained for two, four, and six modalities and compared. (c) Models trained on full six modalities in EPQ-AQS are compared. We select six representative models for the illustration. All values are RMSE in physical units.

to limited spatial coverage, *Interpolation* records slightly better. The superior performance of both *Mid-Fusion* and *ICMR* promotes the values of incorporating modality-specific sensors into model training.

## 5.3 BASELINE COMPARISONS ON MULTIMODAL FORECASTING

Having established in Section 5.2 that multimodality helps under sparse, partly co-located sensing, we now demonstrate how OmniField compares to widely used baselines when forecasting real scientific fields from sparse multimodal inputs.

### 5.3.1 COMPARISONS ON CLIMSIM–THW

Our hypothesis is that continuity-informed conditioning with *ICMR* will outperform *Co-Location*, *Interpolation*, or *Mid-Fusion* alternatives by preserving native sparsity while aligning modalities in a shared latent field *prior* to decoding. We use the ClimSim–THW setup from Section 5.1. All baselines are trained with identical splits and normalization (per-modality z-score). Baselines inherit the originally suggested model sizes with additional overhead related to multimodal adaptation. We report per-modality errors over $(T, H, W)$. • **Results:** Table 2 shows that OmniField achieves the best mean error and leads on most per-modality comparisons. Among originally-unimodal baselines, *Mid-Fusion* is stronger than *Co-Location* and *Interpolation*, confirming that retaining native, modality-specific supports matters. Nevertheless, Mid-Fusion remains below OmniField, indicating that exchanging cross-modal context before conditioning a unified field is critical. Across architectures, CNF-style models outperform operator-learning (e.g., FNO/OFormer) and standard CNNs (UNet/ResNet). Natively multimodal designs (e.g., PROSE-FD, MIA) benefit from multiple channels but still trail OmniField, suggesting that continuity-aware conditioning and iterative refinement confer additional gains beyond simple multi-channel fusion. Qualitative comparisons in Fig. 3 illustrate a set of inputs, ground truths, and field forecasting results from six models. Given mere 3.87% sampling rate, OmniField is capable to accurately predict the underlying phenomena.

### 5.3.2 COMPARISONS ON NATURALISTIC EPA-AQS

EPA–AQS stress-tests robustness under day-to-day nonstationarity, irregular station layouts, and asynchronous sensing. We test whether OmniField yields reliable multi-pollutant forecasts from sparse, partially co-located inputs and whether adding modalities improves performance when station supports vary by day. Inputs and targets are taken on their day-specific masks, while no imputation is performed. We evaluate: (i) modal-subset scaling, training/evaluating on $|\mathcal{M}| \in \{2, 4, 6\}$ pollutants—M2 = $\{O_3, PM_{2.5}\}$, M4 adds $\{PM_{10}, NO_2\}$, and M6 adds $\{CO, SO_2\}$—and measuring forecasting error for $O_3$ and $PM_{2.5}$; and (ii) full-modal comparison, training/evaluating all methods on M6 and reporting per-pollutant performance. • **Results:** Fig. 4(b) shows that perfor-

Table 2: Baseline Comparisons for Multimodal Forecasting on ClimSim-THW. Mid-Fusion column denotes *Mid-Fusion* except natively multimodal (denoted in Architecture as MM), in which case respective multimodal fusion method is used. T=Temperature (K); H=Humidity ($10^{-3}\,\mathrm{kg\,kg^{-1}}$); W=Wind speed ($\mathrm{m\,s^{-1}}$); # Params=number of model parameters; All values are RMSE in physical units. (refer to Appendix I on model sizes for EPA-AQS)

| Model | Architecture | # Params | Co-Location | | | Interpolation | | | Mid-Fusion | | |
|---|---|---|---|---|---|---|---|---|---|---|---|
| | | | T | H | W | T | H | W | T | H | W |
| UNET | CNN | 53.1M | 5.31 | 1.96 | 5.63 | 4.60 | 1.84 | 5.52 | 4.49 | 1.74 | 5.41 |
| RESNET | CNN | 1.2M | 9.14 | 3.80 | 5.43 | 8.72 | 3.51 | 5.42 | 8.13 | 3.26 | 5.36 |
| OFORMER | Transformer | 2.1M | 12.56 | 4.99 | 6.02 | 11.27 | 4.20 | 5.98 | 11.20 | 4.24 | 5.84 |
| FNO | Operator | 1.1M | 5.98 | 1.89 | 8.51 | 4.17 | 1.80 | 7.25 | 3.36 | 1.39 | 7.19 |
| CORAL | Operator | 2.0M | 14.19 | 4.98 | 7.95 | 13.64 | 4.97 | 7.02 | 13.12 | 3.80 | 6.76 |
| SCENT | CNF | 29.3M | **1.86** | 1.28 | 5.15 | 1.86 | 1.28 | 5.15 | 1.52 | 0.99 | 5.07 |
| PROSE-FD | Operator (MM) | 16.0M | 5.20 | 1.65 | 5.30 | 5.18 | 1.65 | 5.30 | 5.20 | 1.65 | 5.30 |
| MIA | CNF (MM) | 0.3M | 4.63 | 1.74 | 5.26 | 4.69 | 1.61 | 5.19 | 4.43 | 1.63 | 5.26 |
| OmniField | CNF (MM) | 37.4M | 1.93 | **1.00** | **5.07** | **1.40** | **0.81** | **4.97** | **1.07** | **0.66** | **4.86** |

Table 3: Ablation of Our Architecture. ✓indicates activated modules. CIFAR-10 shows MSE, and ClimSim-LHW columns show RMSE in physical units. Promotion is noted next to performance as relative increase in error.

| GFF | Sinusoidal Init. | ICMR | CIFAR-10 (MSE) | ClimSim-LHW (RMSE) | | |
|---|---|---|---|---|---|---|
| | | | | Temperature (K) | Humidity ($10^{-3}\mathrm{kg\,kg^{-1}}$) | Wind ($\mathrm{m\,s^{-1}}$) |
| ✓ | ✓ | ✓ | **0.0007** ($\times 1.00$) | **1.07** ($\times 1.00$) | **0.66** ($\times 1.00$) | **4.86** ($\times 1.00$) |
| ✗ | ✓ | ✓ | 0.0097 ($\times 13.86$) | 2.61 ($\times 2.44$) | 1.74 ($\times 2.62$) | 5.35 ($\times 1.10$) |
| ✓ | ✗ | ✓ | 0.0011 ($\times 1.57$) | 1.08 ($\times 1.01$) | 0.71 ($\times 1.07$) | 4.87 ($\times 1.00$) |
| ✓ | ✓ | ✗ | 0.0053 ($\times 7.57$) | 1.56 ($\times 1.45$) | 0.82 ($\times 1.24$) | 4.91 ($\times 1.01$) |
| ✗ | ✗ | ✓ | 0.0098 ($\times 14.00$) | 1.79 ($\times 1.67$) | 1.30 ($\times 1.96$) | 5.24 ($\times 1.08$) |
| ✗ | ✗ | ✗ | 0.0145 ($\times 20.71$) | 2.92 ($\times 2.72$) | 1.86 ($\times 2.81$) | 5.37 ($\times 1.11$) |

mance scales *monotonically* from M2 → M4 → M6 for most methods, with OmniField leading at each level—evidence that more modalities help even when station supports differ daily. In the full-modal comparison (Fig. 4(c)), some baselines display strengths on particular pollutants; nevertheless, OmniField attains the overall state of the art across pollutants.

## 5.4 Ablations

We isolate which architectural choices drive gains across regimes of increasing difficulty: (i) purely spatial grids (CIFAR-10), (ii) spatiotemporal grids (RainNet), and (iii) sparse, multimodal spatiotemporal point-clouds (ClimSim–THW). We ablate three components introduced for CNF: (a) GFF for query and token embeddings; (b) sinusoidal initialization for cascaded latents; and (c) ICMR (replacing it with a Mid-Fusion fallback). Each toggle changes one component while keeping others fixed. • **Results:** Table 3 summarizes three consistent trends. (i) On **CIFAR-10** reconstruction, combining *GFF + sinusoidal init + ICMR* records the best performance, substantially improving high-frequency recovery over any partial variant. (Appendix E) (ii) On **RainNet** (unimodal spatiotemporal), our architectural advances helps outperform both SCENT and the RainNet baselineS (Fig. 5(d)). (iii) On **ClimSim–THW**, each component helps, and the full model consistently reduces error across $T$, $H$, and $W$. Overall, the ablations support our design choice: frequency-rich embeddings plus a continuity-aware, iteratively refined latent are necessary to perform well across spatial, spatiotemporal, and multimodal regimes.

## 5.5 Robustness to Noisy Sensors

Observational networks exhibit outliers, dropouts, and calibration drift. A desirable property is robustness to corrupted inputs—ideally the model should route information through cleaner channels

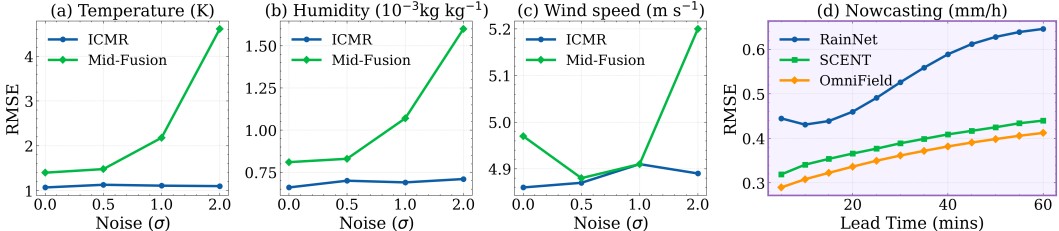

Figure 5: (a)-(c) *ICMR* is contrasted against *Mid-Fusion* in an increasing amount of instance-level noise severity. (d) Our OmniField outperforms both SCENT and RainNet on an established rainfall nowcasting task.

and suppress noisy ones. On ClimSim–THW, we compare *ICMR* versus *Mid-Fusion* under increasing degree of noise corruption. For each sample, we uniformly choose $k \in \{1, 2\}$ modalities from $\{T, H, W\}$ and add Gaussian noise with standard deviation $\sigma \in \{0.5, 1.0, 2.0\}$ scaled by the sample's observed input standard deviation for that modality. At most two modalities are corrupted per sample, so at least one remains clean. Evaluation uses clean inputs and targets. • **Results:** Fig. 5(a)-(c) shows a clear advantage on robustness: *ICMR* remains near its clean-input accuracy across all corruption scales, whereas *Mid-Fusion* deteriorates steadily as noise increases. At the highest corruption level, *Mid-Fusion* incurs large errors on temperature and humidity and a noticeable drift on wind, while *ICMR* remains virtually unchanged on all three targets. Even under mild noise, *ICMR*'s metrics move only marginally, indicating that its iterative cross-modal refinement routes information through cleaner channels and suppresses corrupted ones. In contrast, Mid-Fusion—lacking this pre-conditioning exchange—amplifies noise introduced in any single modality and propagates it to the shared representation. For wind, Mid-Fusion shows a small error dip at low $\sigma$: a noise-as-regularization effect where tiny zero-mean perturbations during training discourage overfitting to idiosyncratic station readings and slightly improve clean-test error; the gain vanishes and reverses as $\sigma$ increases.

## 5.6 ADDITIONAL EXPERIMENTS

We extend our evaluation with compute/scale measurements, long-horizon rollouts, and downstream transfers. Overall, OmniField sits on a favorable accuracy–efficiency frontier, stays stable up to 48 h unrolled horizons, and learns representations that transfer across modalities and tasks (Appendix Tables 4-8). Beyond aggregate scores, these experiments probe three deployment-relevant questions—*how much compute is needed*, *whether errors compound under rollout*, and *whether learned fields remain useful off-distribution*—which are central for practical use.

**(A) Efficiency and scalability.** We report ClimSim-THW forecasting RMSE together with parameter counts and forward-pass FLOPs (Appendix 4). At an approximately matched budget ($\sim$10M parameters), OmniField substantially improves over FNO (e.g., T RMSE 1.10 vs. 2.96) while remaining far cheaper than PROSE-FD in FLOPs. The benefit is not capacity-only: a 1.3M-parameter OmniField variant already outperforms a much larger FNO configuration (Appendix 4), and scaling OmniField further continues to improve accuracy. Notably, these trends hold consistently across T/H/W rather than being driven by a single variable, suggesting the gains come from the architecture's multimodal field coupling rather than metric-specific effects. This positions OmniField as a practical option when both accuracy and inference cost matter, e.g., for high-frequency forecasting or large ensembles.

**ICMR iteration sensitivity.** Sweeping refinement stages $L \in \{1, 3, 5, 10\}$ (Appendix 5) shows accuracy improves with more refinement; for example, moving from $L=1$ to $L=10$ reduces THW RMSE from 1.383/0.905/5.289 to 0.95/0.60/4.83, with diminishing returns beyond $L \approx 5$. Importantly, the FLOPs increase sublinearly relative to the error reduction, indicating that refinement is an efficient way to allocate capacity within a fixed backbone. In practice, this sweep motivates selecting $L$ as a tunable knob: smaller $L$ for strict latency budgets and moderate $L$ for accuracy-critical settings.

**(B) Long-horizon forecasting stability.** We train OmniField using either 6 h or 48 h supervised horizons and evaluate 6/24/48 h rollouts via normalized RMSE (Appendix 6). With 48 h training,

OmniField remains stable up to 48 h (e.g., $0.063/0.095/0.754$ at 48 h), while training only at 6 h degrades more at long rollouts. In contrast, naive FNO unrolling becomes unstable and diverges at longer horizons, consistent with compounding error; this supports our warp-unrolling design Park et al. (2025). The gap between 6 h- and 48 h-trained OmniField suggests that long-horizon supervision is not merely improving one-step accuracy, but calibrating the dynamics under repeated application. This stability pattern suggests that reference-aligned prediction helps prevent systematic drift, which is a common failure mode in long rollouts..

**(C) Downstream generalization across tasks.** OmniField's learned field representations transfer beyond the primary regression setting (Appendix 7). On CESNET Koumar et al. (2025), OmniField improves over a strong baseline (MSE $0.755$ vs. $0.859$ for TimeMixer Wang et al. (2024)). On sparse-pixel CIFAR-10 (20% observed), fine-tuning reaches $71.8\%$ accuracy. On SEVIR Veillette et al. (2020) anomaly detection, OmniField reaches ROC–AUC $0.97$ (vs. $0.96$ for an FNO baseline). These results span distinct input structures (irregular sparse pixels, multivariate sequences, and spatiotemporal radar imagery), suggesting that the continuous-field parameterization learns reusable primitives rather than dataset-specific shortcuts. Moreover, the competitiveness on classification and detection indicates that the learned features are not limited to reconstruction, but also support discriminative objectives after minimal adaptation.

**(D) Randomly missing modalities.** Randomly dropping one or two modalities during training for a fraction $p$ of instances has limited impact (Appendix 8): at $p=0.5$, OmniField remains close to full-modality training (e.g., THW RMSE $1.11/0.71/4.89$ vs. $1.07/0.66/4.86$ at $p=0$), indicating robustness to intermittent modality absence. This robustness is useful in realistic sensing pipelines where channels can be delayed, corrupted, or unavailable without requiring specialized retraining for each failure pattern. The mild degradation also suggests OmniField can implicitly compensate via cross-modal field consistency, rather than relying on any single modality.

## 6 DISCUSSION AND CONCLUSION

OmniField addresses the two practical obstacles introduced in Section 1—irregular, noisy measurements and variable, partially co-located modality coverage—by learning a single continuity-aware neural field and repeatedly exchanging cross-modal context before decoding. The multi-modal crosstalk (MCT) module and iterative cross-modal refinement (ICMR) align per-modality tokens with a lightweight global code, while fleximodal fusion carries presence/quality masks so the same model operates on any subset of inputs. This design unifies reconstruction, spatial interpolation, forecasting, and cross-modal prediction without gridding or imputation, and empirically yields robust gains: multimodality consistently helps under sparse sensing (Fig. 4(a)), OmniField outperforms strong baselines on ClimSim-THW (Table 2), scales with added modalities on EPA-AQS (Fig. 4(b)), and remains stable under missing/irregular inputs (Fig. 5).

Limitations remain. Compute and memory scale with the number of tokens (e.g., stations) and latent capacity, which can challenge extreme-scale deployments; the current decoder provides point estimates without calibrated uncertainty; generalization under domain shift (seasonal changes, sensor re-sitings, policy-driven coverage changes) is not yet fully quantified; and very long forecasting horizons beyond those studied here may require additional temporal structure.

These gaps suggest clear next steps. Efficiency can be improved via adaptive tokenization/pruning and model distillation (potentially with mixture-of-experts routing); reliability via uncertainty-aware objectives (e.g., quantile or heteroscedastic losses) and ensembling; and robustness via continual/domain-adaptive training. For longer horizons, hierarchical temporal warping or hybrid dynamical priors are natural extensions. Finally, we plan to incorporate and evaluate OmniField in broader scientific settings where multimodal, irregular sensing is ubiquitous—climate reanalysis/forecasting, air-quality operations, materials characterization, particle physics, and multimodal bioimaging—leveraging the fleximodal design to handle missing or asynchronous channels in real deployments.

ACKNOWLEDGEMENTS

This work was supported by the U.S. Department of Energy (DOE), Office of Science (SC), Advanced Scientific Computing Research program under award DE-SC-0012704 and used resources of the National Energy Research Scientific Computing Center, a DOE Office of Science User Facility using NERSC award NERSC DDR-ERCAP0030592.

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

## A  DATA STATISTICS AND HYPERPARAMETERS

We report a detailed training and data specifications for better reproducibility. Data statistics and hyperparameters for OmniField on ClimSim-THW and EPA-AQS.

| | DATASET NAME | |
|---|---|---|
| | **ClimSim-THW** | **EPA-AQS** |
| **DATA STATISTICS** | | |
| Spatial domain / resolution | 1 level, $N = 21600$ grid pts | 1048 fixed sites |
| Modalities / targets | $T, H, W \rightarrow T, H, W$ | $O_3$, $PM_{2.5}$, $PM_{10}$, $NO_2$, CO, $SO_2 \rightarrow$ same six |
| Train / Val size | 9000 / 1000 | time-aware split: 80% train / 20% val (by day) |
| Temporal horizon | random $\tau \in \{1\text{–}6\}$ hr | lead steps $\in \{1\text{–}5\}$ |
| Input sparsity (per sample) | 0.02 ( 432 of 21600); region=union | irregular (observations vary per day/site) |
| Input points per modality | variable (T,H,W sampled at 2%) | variable; min sites to sample per batch: $O_3$: 20, $PM_{2.5}$: 20, $PM_{10}$: 15, $NO_2$: 10, CO: 10, $SO_2$: 10 |
| Normalization | per-variable mean/std | per-pollutant mean/std |
| **TRAINING** | | |
| Batch size (train / val) | 8 / 1 | 4 / 4 |
| Total optimizer steps | 100,000 | epoch-based; 30 epochs |
| Optimizer | AdamW ($\beta_1 = 0.9$, $\beta_2 = 0.999$) | AdamW ($\beta_1 = 0.9$, $\beta_2 = 0.999$) |
| Weight decay | $1 \times 10^{-4}$ | $1 \times 10^{-4}$ |
| LR schedule | CosineAnnealing WarmupRestarts | CosineAnnealing WarmupRestarts |
| Max / Min LR | $8 \times 10^{-5}$ / $8 \times 10^{-6}$ | $8 \times 10^{-5}$ / $8 \times 10^{-6}$ |
| Warmup steps | 1000 | 10% of cycle |
| **MODEL** | | |
| # stages ($L$ in *ICMR*) | 3 | 3 |
| Latent dims per stage | $(128, 128, 128)$ | $(64, 64, 64)$ |
| # latents per stage | $(128, 128, 128)$ | $(64, 64, 64)$ |
| Self-attn trunk blocks (final) | 3 (SA+FF each) | 3 (SA+FF each) |
| Cross-attn heads / dim_head | 4 / 128 | 2 / 32 |
| Self-attn heads / dim_head | 8 / 128 | 2 / 32 |
| Feed-forward multiplier | 4 (GEGLU) | 4 (GEGLU) |
| Input projection per modality | $3 \rightarrow 128$ MLP (T/H/W), then concat pos enc | $3 \rightarrow 128$ MLP per pollutant (value, lat, lon) |
| Decoder heads | three heads: $T$, $H$, $W$ (each $[B, N, 1]$) | six heads: $O_3$, $PM_{2.5}$, $PM_{10}$, $NO_2$, CO, $SO_2$ |
| **EMBEDDING / POSITIONAL FEATURES** | | |
| Spatial encoding (GFF) | in:2 $\rightarrow$ 32 bands (sin+cos=64); scale 15.0 | in:2 $\rightarrow$ 32 bands (sin+cos=64); scale 15.0 |
| Time encoding (GFF) | in:1 $\rightarrow$ 16 bands (sin+cos=32); scale 10.0 | in:1 $\rightarrow$ 32 bands (sin+cos=64); scale 15.0 |
| Query dim | 64 (space) + 32 (time) = 96 | 64 (space) + 64 (time) combined by sum $\Rightarrow$ 64 |
| Per-modality input MLP dim | 128 (before concat with pos enc) | 128 (token dim pre-attention) |

# B  OMNIFIELD ENCODER DETAILS

We introduce the following two advances to improve high frequency learning aspects over the prior art, SCENT (Park et al., 2025).

## B.1  GAUSSIAN FOURIER FEATURES

In our architecture, we use Gaussian Fourier features (Tancik et al., 2020) in place of standard sinusoidal Fourier features. This substitution is motivated by the need to capture high-frequency, fine-grained details more effectively. By sampling frequencies from a continuous Gaussian distribution, we provide the model with a richer and less biased spectral representation than a fixed, discrete set of functions allows (Tancik et al., 2020). Our experimental results demonstrate that this approach significantly enhances the model's ability to reconstruct intricate details, leading to improved performance. For a single coordinate variable $x$, GFF is denoted as follows:

$$\gamma(x) = \begin{bmatrix} \cos(2\pi B x) \\ \sin(2\pi B x) \end{bmatrix} \in \mathbb{R}^{2d}, \quad B \in \mathbb{R}^{d \times 1}, \quad B_{ij} \sim \mathcal{N}(0, \sigma^2) \tag{2}$$

## B.2  SINUSOIDAL INITIALIZATION

To better capture high–frequency structure and stabilize training, we seed the $M$ learnable queries with a compact multi-scale sinusoidal pattern. Let the model dimension be $D$ (even) and set $d = D/2$. Define a log-spaced frequency vector

$$\boldsymbol{\nu} = \begin{bmatrix} \nu_0, \dots, \nu_{d-1} \end{bmatrix}^{\top} \in \mathbb{R}^{d \times 1}, \qquad \nu_k = b^{-k/d} \quad (k = 0, \dots, d-1), \ b > 1.$$

Using the same feature map form as in the GFF paragraph, for each $m \in \{0, \dots, M-1\}$ set

$$q_m^{(0)} = s \begin{bmatrix} \cos(2\pi \boldsymbol{\nu} m) \\ \sin(2\pi \boldsymbol{\nu} m) \end{bmatrix} \in \mathbb{R}^{2d} = \mathbb{R}^D, \qquad Q^{(0)} = [\, q_0^{(0)}; \dots; q_{M-1}^{(0)} \,] \in \mathbb{R}^{M \times D}.$$

Since $\sin^2 + \cos^2 = 1$ elementwise,

$$\|q_m^{(0)}\|_2^2 = s^2 \sum_{k=0}^{d-1} \Big( \sin^2(2\pi \nu_k m) + \cos^2(2\pi \nu_k m) \Big) = s^2 d,$$

so $s = d^{-1/2}$ makes each row roughly unit norm, keeping initial attention logits well scaled.

## C  FLEXIMODAL FUSION

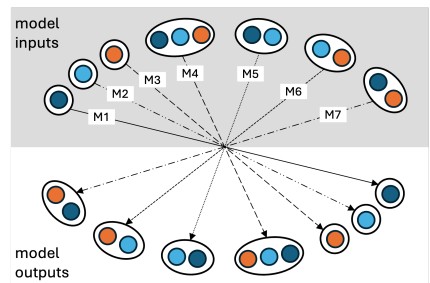
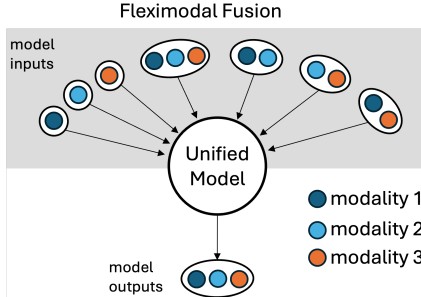

Let $\mathcal{M} = \{1, \ldots, M\}$ index modalities. For instance, let $X_m = \{(\mathbf{x}_i, v_i)\}_{i=1}^{N_m}$ be the tokenized inputs of modality $m$ (possibly empty), and let $\boldsymbol{\pi} \in \{0, 1\}^M$ indicate observed modalities ($\pi_m = 1$ iff $N_m > 0$). Fleximodal fusion learns a single model that (i) conditions on any subset of modalities via presence masks, (ii) gates cross-modal interactions so absent channels contribute neither signal nor noise, and (iii) defines an objective valid for any subset of inputs/targets.

### C.1  MASKED ENCODERS AND GATED FUSION.

Each modality has an encoder $\mathcal{E}_m$ that maps tokens to a latent set $Z_m = \mathcal{E}_m(X_m)$. We gate absent channels at the source:

$$\tilde{Z}_m = \pi_m \mathcal{E}_m(X_m) \qquad (\tilde{Z}_m = \mathbf{0} \text{ when } \pi_m = 0). \tag{3}$$

A fusion operator $\mathcal{A}$ (our MCT/ICMR stack) aggregates only *present* latents to produce the conditioned field representation $G$:

$$G = \mathcal{A}(\{\tilde{Z}_m\}_{m \in \mathcal{M}}), \tag{4}$$

with attention masks that zero cross-attention to absent sets. Concretely, for a cross-attention block with query $Q$ and keys/values $K_m, V_m$ from modality $m$,

$$\text{Attn}(Q, K, V) = \text{softmax}\left(\frac{QK^\top}{\sqrt{d}} + \log M\right)V, \quad M = \text{diag}\left(\underbrace{1, \ldots, 1}_{\text{present}}, \underbrace{-\infty, \ldots, -\infty}_{\text{absent}}\right), \tag{5}$$

so keys/values from $m$ are effectively removed whenever $\pi_m = 0$.

**Query decoding and flexi-objective.**  Given query set $Q$ (locations/times), per-modality decoders $\mathcal{D}_m$ read from $G$ to produce predictions $\hat{Y}_m = \mathcal{D}_m(G; Q)$. Let $\boldsymbol{\tau} \in \{0, 1\}^M$ indicate which targets are supervised on this instance (e.g., modality present on the target day). The loss is defined for *any* input/target subset:

$$\mathcal{L} = \sum_{m=1}^{M} \tau_m \ell(\hat{Y}_m, Y_m), \tag{6}$$

with $\ell$ the per-modality error (we use MSE in normalized space). This objective works for full, partial, or single-modality inputs, and any supervised set of outputs, without imputing missing channels.

**Why this matters.**  Scientific systems frequently yield $\pi_m = 0$ for some $m$ (e.g., QA-filtered satellite AOD, down $PM_{2.5}$ stations, asynchronous sampling). Fleximodal fusion prevents "hallucinated" evidence from absent channels and lets the same trained model operate across variable sensing conditions. This aligns with recent work on missing-modality learning and FuseMoE for fleximodal fusion Han et al. (2024); Wu et al. (2024), and differs from multimodal dropout Neverova et al. (2015), which randomly drops modalities as a training-time augmentation rather than handling truly absent channels.

# D    FURTHER PREPROCESSING DETAILS ON CLIMSIM

**Source and configuration.**    We use the *High-Resolution Real Geography* configuration of Clim-Sim (E3SM-MMF) at $1.5° \times 1.5°$ horizontal resolution (21,600 grid columns), saved every 20 minutes for 10 simulated years. Data are hosted on Hugging Face under `LEAP/ClimSim_high-res`. Each 20-minute step provides an input (mli) and output (mlo) NetCDF pair.

**Modalities (T,H,W).**    We select three prognostic variables at a single vertical level (0-indexed index 59): air temperature `state_t`, specific humidity `state_q0001`, and meridional wind `state_v`. The latitude/longitude for each column are taken from `ClimSim_high-res_grid-info.nc`.

**Download and subsample.**    We downloaded the first 10,000 input snapshots (20-minute spacing) from the training split using a scripted fetch from: `https://huggingface.co/datasets/LEAP/ClimSim_high-res/resolve/main/train/<YYYY-MM>/E3SM-MMF.mli.<YYYY-MM-DD>-<SSSSS>.nc`

For each file, we extract `state_t[state_level=59]`, `state_q0001[state_level=59]`, and `state_v[state_level=59]` into compressed `.npz` files (one snapshot per file). We use a train/test split of 9,000 / 1,000 snapshots.

**Normalization and mesh.**    We precomputed per-modality normalization statistics (mean and standard deviation) over the selected snapshots and applied them consistently across the split. All experiments use the provided high-resolution grid metadata to map each column to $(\varphi, \lambda)$.

Table: ClimSim (High-Res, Real Geography) THW subset used in our experiments.

| QUANTITY | SYMBOL | VALUE |
|---|---|---|
| HORIZONTAL GRID | – | $1.5° \times 1.5°$ (21,600 COLUMNS) |
| TEMPORAL RESOLUTION | $\Delta t$ | 20 MINUTES |
| SNAPSHOTS USED | $N$ | 10,000 (FIRST IN TRAIN) |
| SPLIT (TRAIN/TEST) | – | 9,000 / 1,000 |
| MODALITIES | – | T=STATE_T, H=STATE_Q0001, W=STATE_V |
| VERTICAL LEVEL | $k$ | INDEX 59 (0-INDEXED) |
| GRID METADATA | – | CLIMSIM_HIGH-RES_GRID-INFO.NC |
| NORMALIZATION | – | PER-MODALITY $(\mu, \sigma)$ OVER SELECTED SNAPSHOTS |
| STORAGE | – | ONE SNAPSHOT PER .NPZ (T,H,W AT $k$=59) |

**Preprocessing for Simulating Multimodal Sensor Networks**    When we sparsify each multimodal instance of the ClimSim-THW dataset, we aimed to (i) have all three modalities to share a fixed number of sensors, while (ii) they each have a fixed number of exclusive sensors, and (iii) also a fixed number of paired two-modality sensors. This sophisticated design allows us to simulate a real scenario: where in the scientific settings, data is rarely available for all sensor locations, and it is more naturalistic to have partially shared sensors. Let $X \subset \mathbb{R}^d$ be the continuous spatial domain and let

$$\mathcal{S} = \{x_j\}_{j=1}^{21600} \subset X$$

denote the fixed catalogue of candidate sensor coordinates. For three modalities $m \in \{1, 2, 3\}$, define fixed (instance-invariant) modality-specific sensor sets

$$\mathcal{S}_m \subset \mathcal{S}, \qquad |\mathcal{S}_m| = 0.01\,|\mathcal{S}| = 216.$$

Let the triple co-located subset be

$$\mathcal{S}_\cap := \mathcal{S}_1 \cap \mathcal{S}_2 \cap \mathcal{S}_3, \qquad |\mathcal{S}_\cap| = 0.005\,|\mathcal{S}| = 108.$$

Pairwise overlaps $|\mathcal{S}_i \cap \mathcal{S}_j|$ are otherwise unconstrained (beyond $|\mathcal{S}_i \cap \mathcal{S}_j| \geq |\mathcal{S}_\cap|$), so the covered union satisfies

$$\left|\mathcal{S}_1 \cup \mathcal{S}_2 \cup \mathcal{S}_3\right| = \sum_{m=1}^{3} |\mathcal{S}_m| - \sum_{1 \leq i < j \leq 3} |\mathcal{S}_i \cap \mathcal{S}_j| + |\mathcal{S}_\cap| \in [324, 432],$$

i.e., between $1.5\%$ and $2\%$ of the 21,600 locations.

For each data instance $b \in \{1, \ldots, 10^4\}$ and an input time $t_{\text{in},b}$, the observations available to modality $m$ are the values at its fixed sensor coordinates:

$$U_{t_{\text{in}},b}^{(m)} = \left\{ u_{t_{\text{in}},b}(x) \right\}_{x \in \mathcal{S}_m}, \qquad \text{with } \mathcal{S}_m \text{ independent of } b.$$

Thus the sensor masks $\{\mathcal{S}_m\}_{m=1}^3$ are identical across all $10^4$ instances (and across any forecast targets).

Define the fixed incidence matrix $A \in \{0, 1\}^{21600 \times 3}$ by

$$A_{j,m} = \mathbf{1}\{x_j \in \mathcal{S}_m\}.$$

Then each column sum is $\sum_{j=1}^{21600} A_{j,m} = 216$ and exactly 108 rows satisfy $(A_{j,1}, A_{j,2}, A_{j,3}) = (1, 1, 1)$. Row sums encode co-location patterns $(0, 1, 2, 3$ present at a site$)$, and the masks are fixed over all instances.

# E  PRELIMINARY TEST 1: HIGH-FREQUENCY LEARNING ON CIFAR-10

**Core Hypothesis**  The low frequency bias prevalent in prior works (Park et al., 2025) can be improved by re-designing the two major components of the encoder: positional encoding and learnable queries.

We show below the qualitative generation comparisons, ablating on Gaussian Fourier Features (GFF), Sinusoidal Initialization (SI), and Iterative Cross-Model Refinement (ICMR). Here, since it is a reconstruction task, the time encoding is always zero: $t_h = 0$, and ICMR applies as a hierarchical encoding structure without multimodal mixing. We show eight test data instances for the comparison.

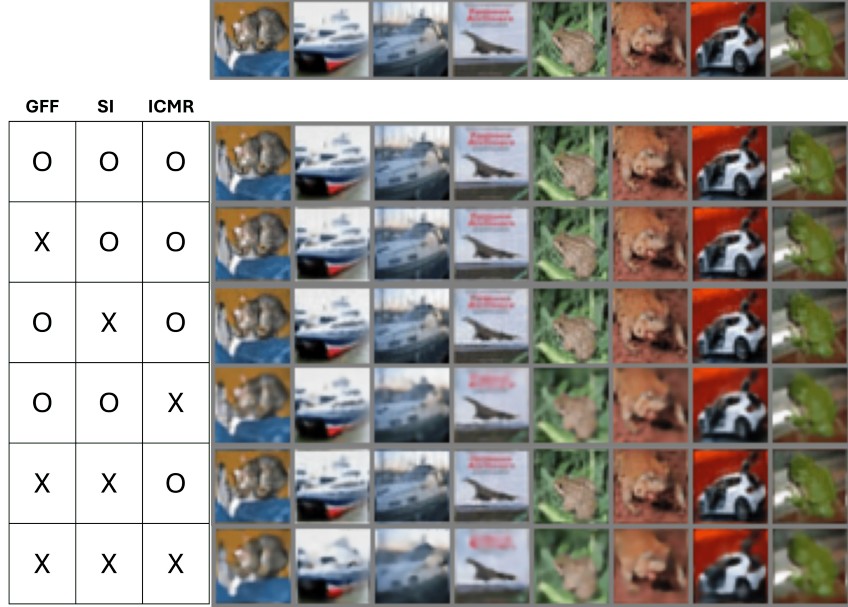

Since the low resolution images of CIFAR-10 makes it hard to compare the results distinctively, we report two additional results. First, we show the MSE performance as shown in the main manuscript Table 5.4. Second, we show the power of the 2D fourier spectrum as below:

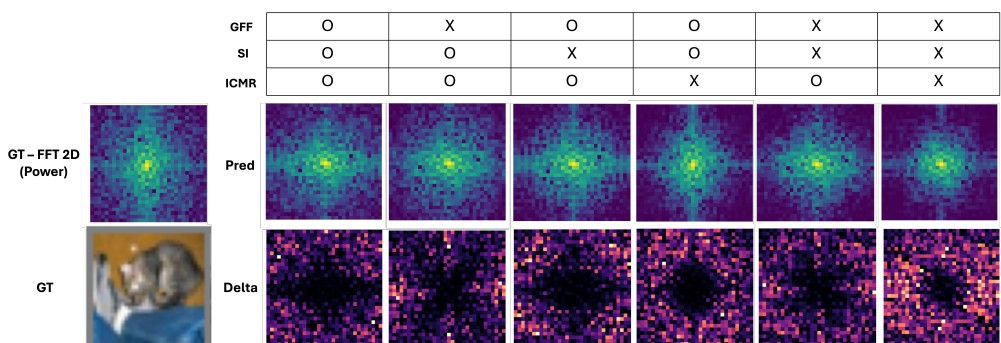

On a test instance, we show the original $32 \times 32$ image, and its corresponding power spectrum as defined by $f = |FFT_{\text{shift}}(FFT(x))|$ where $FFT$ is a 2-dimensional fast fourier transform and $x$ is the input image. The *Delta* is the absolute difference between the ground truth power and the predicted power in the 2D grid. In the *Pred* spectrum, the center area is low-frequency, whereas boundaries are high-frequency components. We see a clear distinction between different model ablations. It is notable that all three components show characteristic impacts on the power spectrum, while the absence of each component leading to a inferior high frequency learning.

## F PRELIMINARY TEST 2: SYNTHETIC SINUSOIDAL MULTIMODAL DATA

**Core Hypothesis.** Differential performance of multimodal CNFs is more easily and directly observable in a toy, synthetic multimodal data.

We constructed a paired spatiotemporal dataset $(S_1, S_2)$ to test conjectures about multimodality in a fully controlled environment. The driving modality $S_1$ is a multiscale sinusoidal field, while the coupled modality $S_2$ evolves under Kuramoto-style phase interactions (Kuramoto, 2005) with $S_1$. This synthetic testbed allows us to isolate the effects of fusion, cross-modal learning, and sparsity.

**Driving modality ($S_1$).** Defined as a sum of sinusoidal components with different spatial/temporal frequencies:

$$S_1(x, t) = \sum_{i=1}^{N} A_i \sin(k_i x - \omega_i t + \phi_i) + \eta(x, t),$$

with $N = 3$, domain $x \in [0, 10]$ discretized to $X = 100$, and $t \in [0, 50]$ discretized to $T = 500$. Noise $\eta$ was not included in the experiments reported here.

**Coupled modality ($S_2$).** Phases $\theta_j$ of $M = 2$ oscillators evolve as

$$\frac{d\theta_j}{dt} = \omega'_j + \frac{K}{N} \sum_{i=1}^{N} \sin(\psi_i(x, t) - \theta_j(t)), \qquad S_2(x, t) = \sum_{j=1}^{M} B_j \sin(\theta_j(t)),$$

where $\psi_i$ are the $S_1$ component phases. Coupling strength $K$ tunes dependency: weak $K$ yields nearly independent signals; strong $K$ synchronizes $S_2$ to $S_1$. Unless otherwise stated, $K = 2.5$.

**Windows and splits.** We form windows with input length $L_{\text{in}} = 20$ and horizon $L_{\text{pred}} = 1$, stride $s = 1$, producing $N_{\text{win}} = 479$ examples. Unless noted, we use an 80%/20% train/validation split. Sparsity masks (when used) are random with a kept fraction of $\sim 30$ points, applied either jointly or per-modality. No normalization was applied.

Table: Synthetic S1/S2 dataset statistics (default configuration).

| QUANTITY | SYMBOL | VALUE |
|---|---|---|
| SPATIAL POINTS | $X$ | 100 |
| TIMESTEPS | $T$ | 500 |
| WINDOW / HORIZON / STRIDE | $(L_{\text{IN}}, L_{\text{PRED}}, s)$ | (20, 1, 1) |
| # WINDOWS | $N_{\text{WIN}}$ | 479 |
| SPLIT (TRAIN/VAL/TEST) | – | 80 / 20 / 0 |
| COUPLING | $K$ | 2.5 |
| NOISE | $\eta$ | NONE |
| SPARSITY | – | OFF / ON ($\sim 50$ POINTS KEPT) |

**Experiment variants.** We evaluated seven complementary configurations to probe information sharing and robustness:

(a) **Multimodal reconstruction:** Input $(S_1, S_2)$, predict both at $t+1$.
(b) **Cross-modal S1→(S1,S2):** Input $S_1$ only, predict both.
(c) **Cross-modal S2→(S1,S2):** Input $S_2$ only, predict both.
(d) **Mixed sparsity:** One modality half dense/half sparse, the other dense.
(e) **Unimodal sparse ($\sim 50$ points):** Input from a single modality, sparsified to $\sim 50$ spatial points.
(f) **Temporal continuity:** Train/evaluate with fractional $t_{\text{out}}$ to test interpolation.

**Summary.** Across variants, multimodal fusion consistently improved reconstruction, forecasting, and cross-modal prediction compared to unimodal baselines, with advantages persisting under sparsity (both multimodal and unimodal at $\sim 50$ points) and fractional time interpolation. These results provide a controlled validation of our multimodal neural field conjectures before applying the method to real scientific datasets.

# G   MORE QUALITATIVE RESULTS ON CLIMSIM-LHW

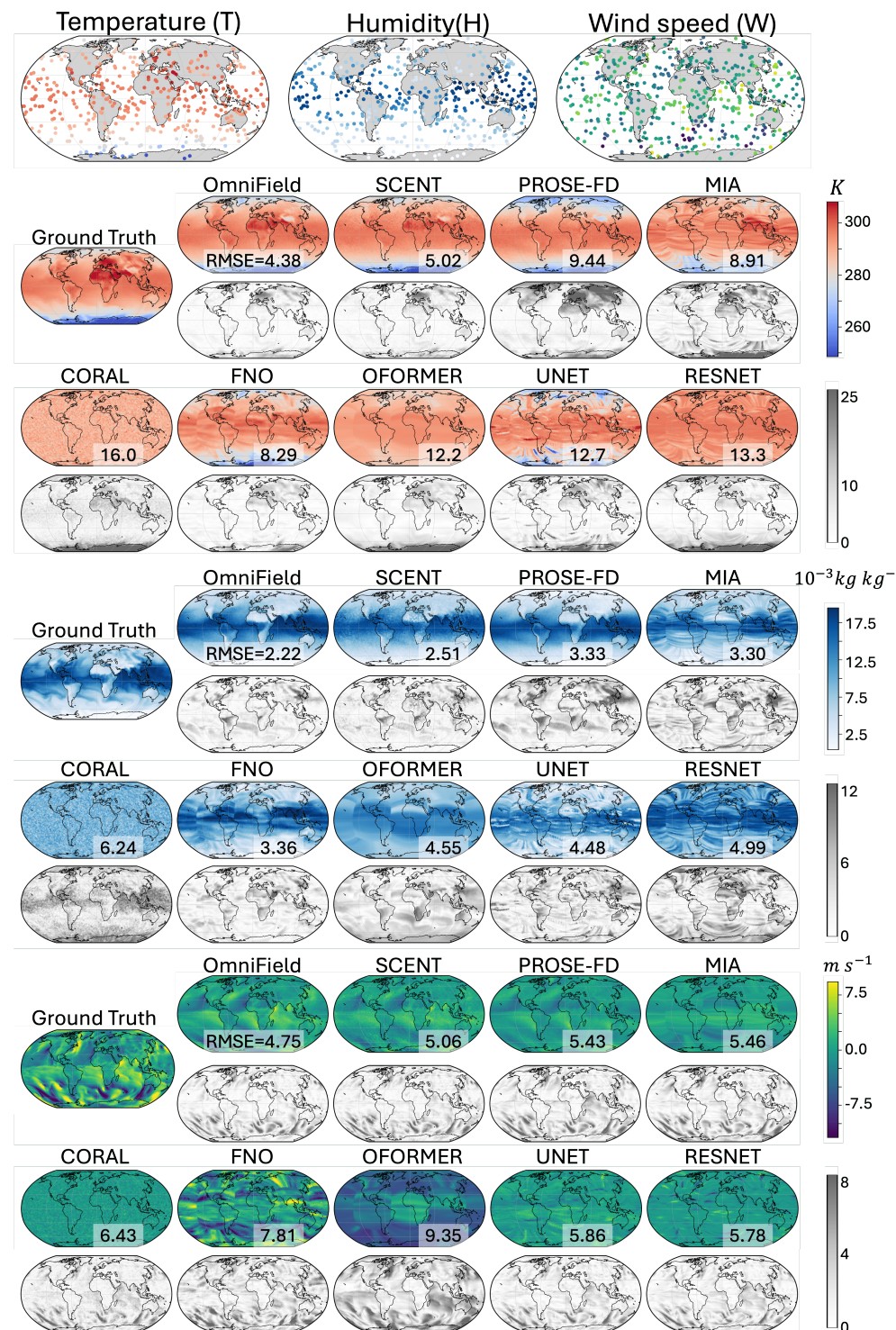

Figure: ClimSim-LHW full qualitative evaluation. Gray-scale images denote the delta in absolute error.

# H    MORE QUALITATIVE RESULTS ON EPA-AQS

Qualitative Comparisons for forecasting ($\Delta t = 5$ days) on sparse EPA-AQS data. Visualized on a subset for six models and three modalities ($O_3$, $PM_{2.5}$, and CO).

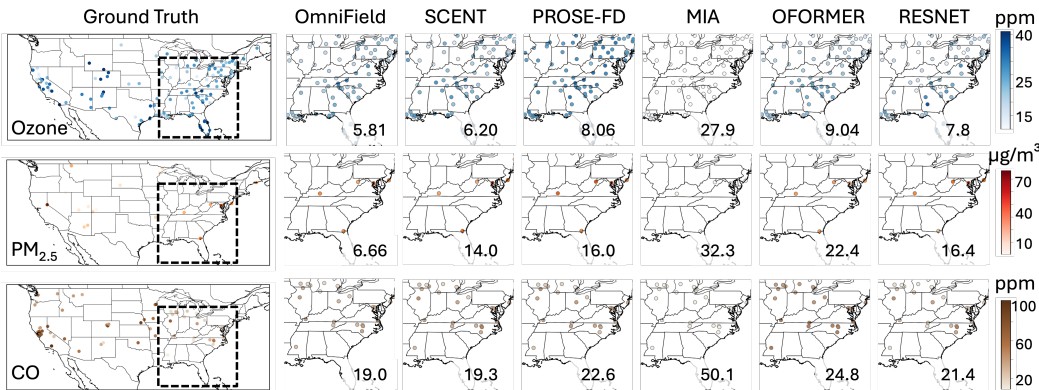

Below we show two uncurated examples of the full multimodal forecasting comparisons on EPA-AQS. The first and second rows correspond to the set of inputs and ground truths, respectively. The input and output have varying sensor locations and statistics, posing significant challenges. Our OmniField's superior performance reveal the proven values of continuity-aware model architecture on top of the proposed ICMR in robust spatiotemporal learning.

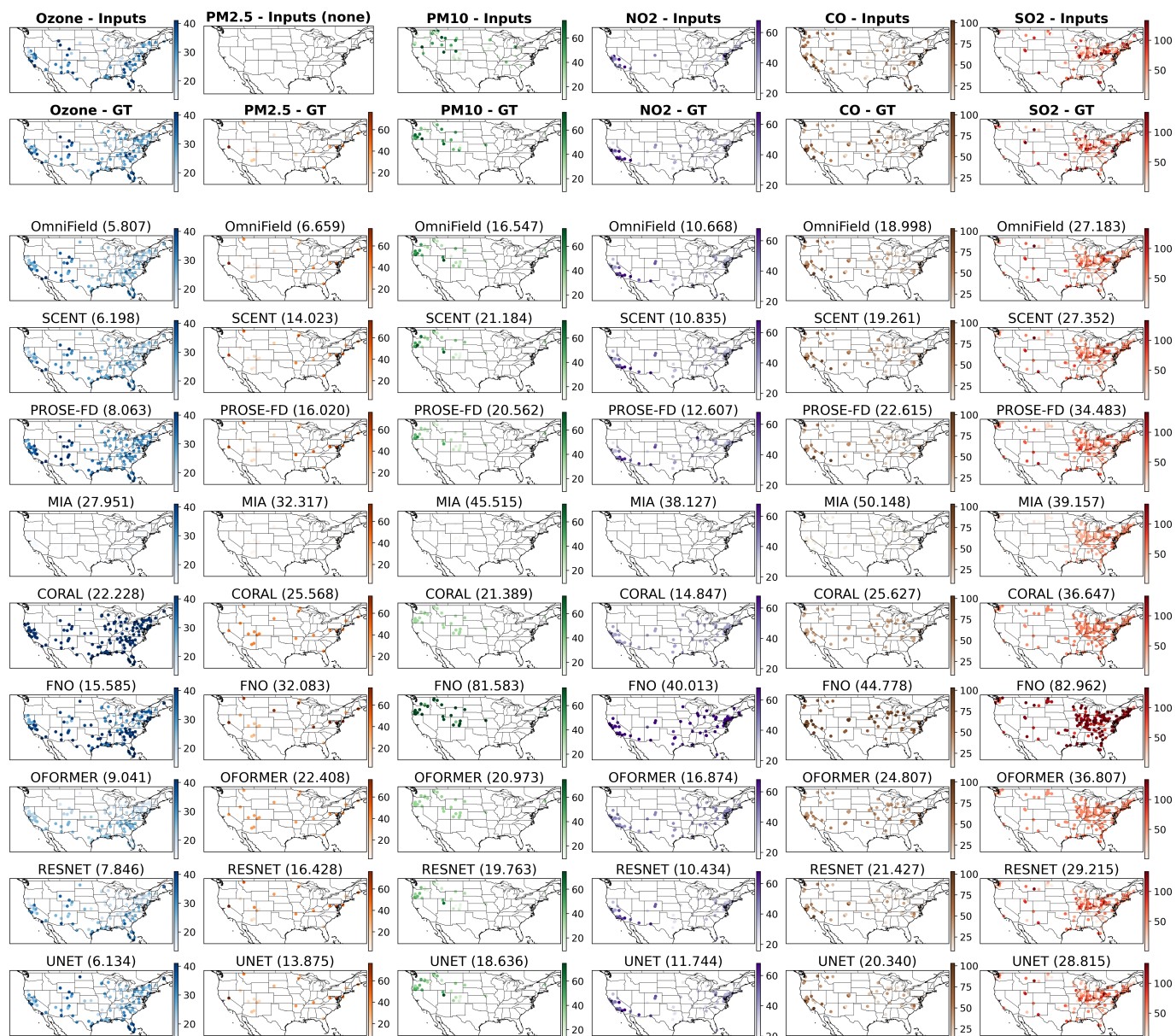

Figure: EPA-AQS full qualitative evaluation - instance 1

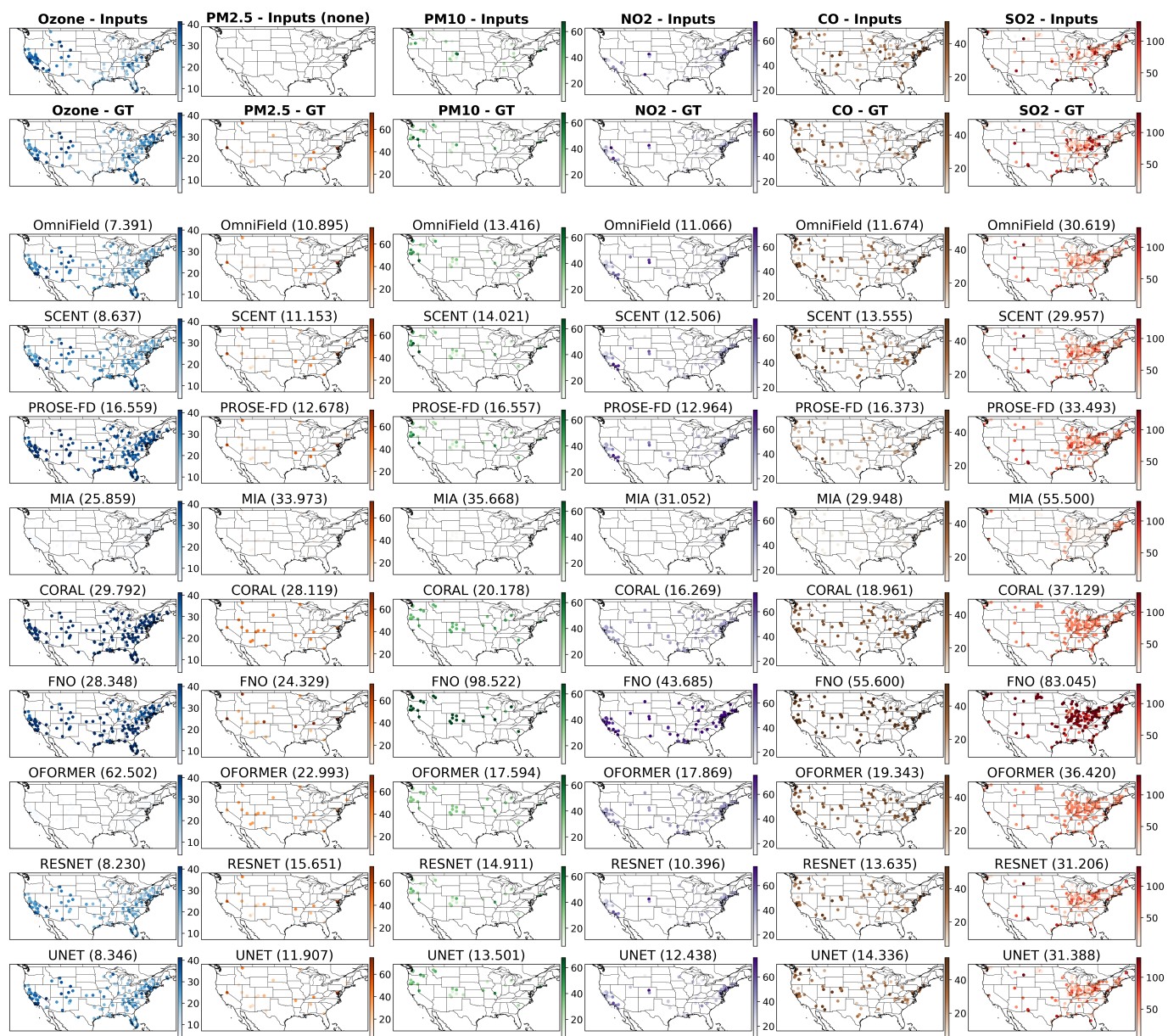

Figure: EPA-AQS full qualitative evaluation - instance 2

## I    FULL EXPERIMENTAL TABLES FOR FIG. 4

Table for Fig. 4(a).  For each input configuration, we show RMSE performance for each output modalities, separately. Note that for Co-Location strategy we limit the test set to the intersection of all sensors from three modalities. In the meanwhile, for Interpolation, Mid-Fusion, or ICMR, we use all sensor locations for the test set. Therefore, we separate the test split and report them separately. Fig. 4(a) corresponds to *Test split = Intersection*. It shows that, regardless of the Test split, ICMR outperforms outer strategies across all modalities. All values are in physical units. Lower is better.

| Strategy | Test split | $\mathcal{M}_{in} = \{T\}$ | | | $\mathcal{M}_{in} = \{T, H, W\}$ | | |
|---|---|---|---|---|---|---|---|
| | | $T$ $[\mathrm{K}]$ | $H$ $[10^{-3}\,\mathrm{kg\,kg^{-1}}]$ | $W$ $[\mathrm{m\,s^{-1}}]$ | $T$ $[\mathrm{K}]$ | $H$ $[10^{-3}\,\mathrm{kg\,kg^{-1}}]$ | $W$ $[\mathrm{m\,s^{-1}}]$ |
| Interpolation | | 2.677 | 1.171 | 5.172 | 1.936 | 1.006 | 5.063 |
| Mid-Fusion | Union | 1.887 | 1.024 | 5.155 | 1.398 | 0.807 | 4.967 |
| ICMR | | **1.530** | **0.870** | **5.011** | **1.073** | **0.664** | **4.859** |
| Co-Location | | 3.093 | 1.315 | 5.265 | 2.527 | 1.157 | 5.137 |
| Interpolation | Intersection | 2.682 | 1.167 | 4.513 | 1.972 | 1.007 | 4.375 |
| Mid-Fusion | | 1.854 | 0.943 | 4.502 | 1.386 | 0.795 | 4.340 |
| ICMR | | **1.655** | **0.841** | **4.380** | **1.081** | **0.645** | **4.258** |

Table for Fig. 4(b–c).  Comparison of models on the EPA-AQS dataset with varying number of training modalities. Test set is always 2 modalities.

| Model | # Params | 2 Modality Train | 4 Modality Train | | 6 Modality Train | | |
|---|---|---|---|---|---|---|---|
| | | $O_3/PM_{2.5}$ | $O_3/PM_{2.5}$ | $PM_{10}/NO_2$ | $O_3/PM_{2.5}$ | $PM_{10}/NO_2$ | $CO/SO2$ |
| OmniField | 4.3M | 4.6/16.4 | 4.5/15.4 | 51.6/8.3 | 4.4/14.9 | 35.3/7.3 | 19.8/26.4 |
| FNO | 0.2M | 11.3/17.6 | 8.7/17.2 | 56.5/9.2 | 8.3/15.6 | 43.9/8.8 | 32.8/31.3 |
| SCENT | 29.3M | 5.7/16.9 | 5.6/16.1 | 54.3/8.5 | 5.5/15.8 | 44.4/7.6 | 25.6/26.7 |
| MIA | 1.5M | 7.9/22 | 8/23.1 | 39.6/14.1 | 7.4/21.1 | 38.8/13.7 | 21.1/42.8 |
| CORAL | 1.9M | 19.1/27 | 9.7/22.2 | 52.7/21.4 | 9.5/21.5 | 77.6/23.1 | 28.5/51.9 |
| PROSE-FD | 42.3M | 5.5/19.3 | 7.9/20.6 | 38.4/7 | 5.4/19.7 | 40.3/12.8 | 19.9/39.4 |
| OFORMER | 2.5M | 9.3/20.6 | 7.7/19.5 | 46.3/15.9 | 7.6/19.4 | 45.9/13.6 | 29.3/38.9 |
| UNET | 13.3M | 7.8/18.3 | 6.5/18 | 46.4/11.4 | 7/17.6 | 45.1/13.2 | 30.7/36 |
| RESNET | 4.5M | 10.2/18 | 8.2/18.5 | 43/11.7 | 8.6/17.9 | 49.4/10.5 | 32.7/34.2 |

## J   ADDITIONAL EXPERIMENTAL TABLES

Table 4: **Efficiency and scalability on ClimSim-THW.** We report forecasting RMSE (physical units) for temperature (T), humidity (H), and wind (W), together with parameter counts and forward-pass FLOPs.

| Model / Setting | Params (M) | FLOPs | T [$K$] | H [$10^{-3}$ kg kg$^{-1}$] | W [m s$^{-1}$] |
|---|---|---|---|---|---|
| OmniField | 1.3 | $1.6\times10^{10}$ | 1.21 | 0.74 | 5.13 |
| OmniField | 10.6 | $3.0\times10^{10}$ | 1.10 | 0.68 | 4.94 |
| OmniField | 37.4 | $3.6\times10^{10}$ | 1.07 | 0.66 | 4.86 |
| FNO | 10.5 | $5.0\times10^{9}$ | 2.96 | 1.16 | 6.79 |
| FNO | 38.1 | $1.7\times10^{11}$ | 2.53 | 1.13 | 6.72 |
| PROSE-FD | 10.6 | $3.1\times10^{12}$ | 5.40 | 1.95 | 5.66 |

Table 5: **ICMR refinement stage sweep.** We vary the number of refinement stages $L \in \{1, 3, 5, 10\}$ and report THW RMSE along with parameters and FLOPs.

| $L$ | Params (M) | FLOPs | T RMSE | H RMSE | W RMSE |
|---|---|---|---|---|---|
| 1 | 5.4 | $2.82\times10^{10}$ | 1.383 | 0.905 | 5.289 |
| 3 | 10.6 | $3.0\times10^{10}$ | 1.10 | 0.68 | 4.94 |
| 5 | 18.5 | $3.2\times10^{10}$ | 0.97 | 0.61 | 4.86 |
| 10 | 34.4 | $3.8\times10^{10}$ | 0.95 | 0.60 | 4.83 |

Table 6: **Long-horizon rollout stability (normalized RMSE).** We train OmniField with either a 6 h or 48 h supervised horizon and evaluate 6/24/48 h rollouts. We also report a naive unrolled FNO baseline. Lower is better.

| Model | Train horizon | Eval horizon | $T$ | $H$ | $W$ |
|---|---|---|---|---|---|
| OmniField | 48 h | 6 h | 0.056 | 0.084 | 0.700 |
| OmniField | 48 h | 24 h | 0.056 | 0.092 | 0.722 |
| OmniField | 48 h | 48 h | 0.063 | 0.095 | 0.754 |
| OmniField | 6 h | 6 h | 0.062 | 0.093 | 0.707 |
| OmniField | 6 h | 24 h | 0.085 | 0.134 | 0.728 |
| OmniField | 6 h | 48 h | 0.087 | 0.138 | 0.789 |
| FNO | – | 6 h | 0.449 | 0.417 | 1.860 |
| FNO | – | 24 h | $2.58 \times 10^{3}$ | $2.66 \times 10^{3}$ | $1.66 \times 10^{4}$ |
| FNO | – | 48 h | $9.36 \times 10^{9}$ | $1.01 \times 10^{10}$ | $6.60 \times 10^{10}$ |

Table 7: **Downstream task transfer.** OmniField representations transfer across tasks and data modalities.

| Task / Dataset | Metric | OmniField | Baselines |
|---|---|---|---|
| CESNET (multivariate TS forecasting) | Test MSE ↓ | 0.755 | TimeMixer: 0.859;   FNO: 0.790 |
| CIFAR-10 (20% sparse pixels) | Acc. ↑ | 71.8% | UNet: 55.0% |
| SEVIR (lightning anomaly detection) | ROC–AUC ↑ | 0.97 | FNO: 0.96 |

Table 8: **Training with randomly missing modalities.** One or two modalities are randomly dropped during training for a fraction $p$ of instances (never dropping all modalities). Reported on ClimSim-THW in physical units.

| Drop fraction $p$ | T RMSE $[K]$ | H RMSE $[10^{-3} \text{ kg kg}^{-1}]$ | W RMSE $[\text{m s}^{-1}]$ |
|---|---|---|---|
| 0.0 | 1.07 | 0.66 | 4.86 |
| 0.5 | 1.11 | 0.71 | 4.89 |
| 0.75 | 1.23 | 0.74 | 4.94 |

