# OpenReview forum: "OmniField: Conditioned Neural Fields for Robust Multimodal Spatiotemporal Learning"
_ICLR.cc/2026/Conference — ICLR 2026 Poster_

### Official Review · Reviewer_BsBj · 2025-10-24

**Soundness:** 2
**Presentation:** 2
**Contribution:** 2
**Rating:** 4
**Confidence:** 3

**Summary:**

This paper proposes a multimodal conditioned neural field (CNF) framework for learning from sparse, irregular, and noisy multimodal spatiotemporal signals, called OmniField. OmniField has three major components: Gaussian Fourier embeddings, sinusoid initialization to stabilize training, and iterative cross-modal refinement to align modalities. The authors evaluate OmniField on both simulated and real-world multimodal geo-spatio-temporal datasets and show notable performance gain over SOTA baselines. Ablation studies are performed on both spatial and spatio-temporal datasets to understand the contribution of each component.

**Strengths:**

- The datasets and benchmarks are comprehensive spanning across multiple applications
- Omnifield shows robustness and performance gains
- Proposed components are validated through ablation studies

**Weaknesses:**

- I am quiet concerned with the novelty. The core framework remains a straightforward extension of SCENT with a few architectural augmentations for multimodal data.
- The authors have limited explanation of training efficiency and scalability. The computational complexity can grow with the number of tokens and modalities, but the paper has limited analysis on the training or inference efficiency, nor does it discuss how OmniField might perform on larger-scale or real-time systems. Given that some spatiotemporal models for some domains are often deployed on resource-constrained platforms, this efficiency is important.
- Although the authors evaluated the methods on numerous datasets, the diversity of the downstream task evaluated is limited. The paper focuses almost entirely on reconstruction and forecasting accuracy. Other tasks such as classification, uncertainty quantification or anomaly detection could enhance the robustness. A stronger connection to practical use cases would also enhance the work’s relevance.
- The model implicitly assumes shared information across modalities but does not explicitly disentangle shared vs. private modality features, which are important in multimodal learning. While the iterative fusion helps align features, it remains unclear how modality-specific noise or private components are handled. Can the authors elaborate on the integration with factorized multimodal learning?
- Minor:
    - More extensive citations in the Introduction, especially from the data/model perspective to contextualize multimodal spatiotemporal learning in prior literature can benefit.
    - What is SCENT-MM? Is that OmniField? Figure 4b and 4c do not have OmniField

**Questions:**

- How sensitive is the performance to the number of refinement iterations? Could fewer iterations achieve comparable results with lower computational cost?
- The fusion mechanism prevents information leakage from missing modalities. Could the authors quantify how performance is under a randomly missing modalities scenario?
- How does OmniField perform in other tasks, such as classification, where the learned field embeddings are fine-tuned for downstream objectives, rather than regression or forecasting?
- How transferable is the approach to non-geospatial multimodal time-series domains (e.g., high-frequency sensor data)? Does the current formulation depend heavily on spatial continuity assumptions?

---

> ### Author Response · Authors · 2025-11-25
>
> **We thank the reviewer for truly constructive feedbacks. Here we show clarifications and comprehensive additional reports to further support our paper.**
>
> **1. Novelty Beyond SCENT**
>
>
> We appreciate the observation that OmniField builds on the continuous neural field paradigm introduced by SCENT. Our goal is indeed to extend that framework into **a practical, robust multimodal system under sparse, irregular, and noisy sensing**, rather than to replace it. At the same time, OmniField introduces **several architectural and algorithmic components** that are not present in SCENT and that we have found to be crucial in real multimodal settings.
> - **First**, OmniField replaces SCENT’s purely unimodal encoder–processor–decoder pipeline with Multimodal Crosstalk (MCT) plus Iterative Cross-Modal Refinement (ICMR). Each modality is encoded into its own token set, and these tokens interact through MCT blocks that repeatedly fuse and refine a shared multimodal representation. Rather than a single fusion pass, we refine the global code across several stages, which empirically **improves robustness to both noise and sparsity**.
> - **Second**, Fleximodal Fusion models the presence or absence of each modality explicitly. At training and inference, we maintain a per-modality mask and gate absent modalities both in the encoders and in cross-attention, preventing any contribution from missing channels. This allows OmniField to **operate with a single model under arbitrary subsets of inputs** (e.g., only T, or T+H, or T+H+W) without retraining or heuristic imputation.
> - **Third**, we adopt a frequency-rich encoder based on Gaussian Fourier Features and sinusoidal initialization for the learnable queries. This mitigates the low-frequency bias typical of standard CNFs and significantly **improves high-frequency reconstruction** in our experiments.
> - **Finally**, we introduce **new multimodal benchmarks and protocols** specifically designed to stress these aspects: sparsified climate fields (ClimSim variants), ML-ready air-quality data with irregular station coverage (EPA-AQS), and multimodal robustness protocols that vary both sparsity and noise.
>
>
> **2. Factorized Multimodal Learning: Shared vs Private Components**
>
>
> We appreciate the reviewer’s pointer to factorized multimodal learning. While we do not parameterize explicit shared/private subspaces, **the per-modality tokens and global latent effectively behave as private vs shared components in practice**; we see this empirically in [e.g., robustness under missing modalities / across subsets].`
>
>
> Each modality is encoded into its **own token set and retains those tokens** (i.e., private component) throughout the network. These tokens capture modality-specific structure, including private information and noise. In each MCT block, we concatenate all modality tokens, jointly process them, and distill their joint structure into a **global latent code** (i.e., global components). This shared code is then broadcast back to all modalities at the next refinement step. Over multiple ICMR iterations, this cycle performs a form of iterative alignment: shared structure is progressively absorbed into the global code, while modality-specific residuals remain in the per-modality tokens.
>
>
> **Fleximodal Fusion further emphasizes this factorization**. Since absent modalities are masked and cannot contribute to the shared code or cross-attention, the model is forced to represent cross-modal invariants in the shared code that remain stable under varying modality subsets, while allowing each modality’s tokens to maintain private variations and noise.
>
>
> In the revision, we will add a short discussion explicitly connecting OmniField to factorized multimodal learning, clarifying how the current design implicitly separates shared and private components, and outlining how more explicit factorized parameterizations (e.g., explicitly allocated shared/private subspaces or mixture-of-experts routing per modality) could be layered on top of our framework in future work.
>
> **3. Robustness to Missing Modalities**
>
>
> To test robustness to **randomly missing modalities**, we train the 37M-parameter model with three ICMR blocks while, for 100·*p*% of training instances, randomly dropping one or two modalities (but never all three). As shown in Table 1, performance is almost unchanged when *p* = 0.5, and even at *p* = 0.75 the degradation is modest (T, H, and W increase by only 0.16, 0.08, and 0.08, respectively, relative to *p* = 0.0), indicating that **the fusion mechanism handles missing inputs with little leaking information from absent modalities**. Finally, note that EPA-AQS naturally has missing modalities both in training and testing, serving as a realistic testbed of missing modalities.
>
>
> **Table 1: Performance under randomly missing modalities**
>
>
> | ***p*** | **T** | **H**  | **W** |
> |---|---|---|---|
> | 0.0 | 1.07 | 0.66 | 4.86 |
> | 0.5 | 1.11 | 0.71 | 4.89 |
> | 0.75 | 1.23 | 0.74 | 4.94 |

---

> ### Author Response · Authors · 2025-11-25
>
> **We agree** that efficiency and scalability are crucial, especially for resource-constrained deployments. Our additional experiments highlight two key points:
>
>
> - **Compute Efficiency**: OmniField is markedly more accurate and consistently achieves a better compute–performance tradeoff than baseline models.
>
>
> - **On Refinement Iterations**: Ablations and a time-complexity analysis show that increasing the number of refinement iterations incurs only a modest increase in compute (FLOPs). Additional iterations monotonically improve performance, but with diminishing returns, clarifying how to choose an efficient operating point.
>
>
> **4.  On Multimodal Efficiency**
>
>
> **Table 2** – Efficiency and accuracy for unimodal and multimodal settings
> |**Model**|**Input mods**|**Params** (M)|**FLOPs** (fwd)|**T** (err)|**H** (err)|**W** (err)|
> |---|---|---|---|---|---|---|
> |FNO|T|10.6|5.0e9|**1.10**|*0.98*|6.48|
> |PROSE-FD|T|10.5|8.8e11|1.28|1.30|*5.76*|
> |OmniField|T|10.6|2.8e10|*1.11*|**0.76**|**5.20**|
> | | | | | | | |
> |FNO|THW|1.1|1.6e9|3.36|1.39|7.19|
> |FNO|THW|10.6|5.0e9|2.96|1.16|6.79|
> |FNO|THW|38.1|1.7e11|2.53|1.13|6.72|
> |PROSE-FD|THW|1.0|3.2e11|5.79|2.31|6.32|
> |PROSE-FD|THW|10.5|3.1e12|5.40|1.95|5.66|
> |PROSE-FD|THW|38.3|6.4e12|3.03|1.18|5.71|
> |OmniField|THW|1.3|1.6e10|1.21|0.74|5.13|
> |OmniField|THW|10.6|3.0e10|*1.10*|*0.68*|*4.94*|
> |OmniField|THW|37.4|3.6e10|**1.07**|**0.66**|**4.86**|
>
>
> We compare OmniField to FNO and PROSE-FD at matched parameter counts for **unimodal** (T) and **multimodal** (THW) settings.
>
>
> **Unimodal (T)**: All models have \~10M parameters, but FNO is cheapest (5.0e9 FLOPs), OmniField is moderately heavier (2.8e10 FLOPs), and PROSE-FD is two orders of magnitude more expensive (8.8e11 FLOPs). OmniField nonetheless attains the best or comparable accuracy, with lower H and W error than both baselines and T error on par with FNO, showing a favorable accuracy–efficiency trade-off against both baselines.
>
>
> **Multimodal (THW)**: At 1M parameters, OmniField already outperforms FNO and PROSE-FD by a wide margin on all variables (e.g., T error 1.21 vs. 3.36 for FNO and 5.79 for PROSE-FD), using ~10× the FLOPs of FNO but >20× fewer FLOPs than PROSE-FD. As capacity scales to 10M and 38M parameters, OmniField continues to improve, reaching T/H/W errors of 1.07/0.66/4.86 at 37.4M parameters with 3.6e10 FLOPs—over an order of magnitude cheaper than 38M PROSE-FD (6.4e12 FLOPs) and cheaper than the largest FNO (1.7e11 FLOPs), while maintaining significantly better accuracy. Unlike FNO and PROSE-FD, where adding modalities can **degrade** performance (e.g., T error 1.10→2.96 when 10M FNO goes from T to THW), OmniField consistently exploits H and W to reduce error without a proportional compute increase.  Overall, at matched capacity, OmniField offers a superior efficiency–accuracy trade-off, particularly in the multimodal regime.
>
>
> **5. On the Number of ICMR Refinements**
>
>
> **Time Complexity Comparisons.** Let $M$ be the number of modalities (e.g., $1 \rightarrow 3 \rightarrow 6$), $D$ the latent/feature width (FNO: channel width $C$; PROSE-FD/OmniField: attention width $d$), and $L$ the number of ICMR refinement iterations (OmniField only).
>
>
> | **Model** | **Big-O** | **Intuition** |
> |---|---|---|
> | FNO | $O(MD)$ | Modalities contribute linearly to channel count; FFT and linear projections scale with width. |
> | PROSE-FD |$O(M^{2}D$) | Single-pass self-attention over fused multimodal tokens. |
> | OmniField | $O(LM^{2}D)$ | Each ICMR stage revisits multimodal fusion; pairwise cross-modal interactions lead to the $M^{2}$ term. |
>
>
> Theoretically, OmniField’s cost grows linearly with $L$. In practice, FLOPs grow *sublinearly* with $L$ because many components (projections, decoders) do not scale with refinement depth. This is evident in the following empirical result.
>
>
> |**ICMR stages**|**Params** (M)|**FLOPs** (forward)|**T** (err)|**H** (err)|**W** (err)|
> |---|---|---|---|---|---|
> |1|5.4|2.82e+10|1.383|0.905|5.289|
> |3|10.6|3.0e+10|1.10|0.68|4.94|
> |5|18.5|3.2e+10|*0.97*|*0.61*|*4.86*|
> |10|34.4|3.8e+10|**0.95**|**0.60**|**4.83**|
>
>
> Scaling $L$ from 1$\rightarrow$10 stages (5.4M$\rightarrow$34.4M params) improves T RMSE 1.383$\rightarrow$0.95 (−31%) with only +35% FLOPs (2.82e10$\rightarrow$3.8e10), whereas widening the model width (Table 2; 10.6M$\rightarrow$37.4M params) yields just 1.10$\rightarrow$1.07 (−2.7%) for +20% FLOPs—highlighting that $L$ is the more effective lever at comparable parameter budgets and supports the pivotal role of ICMR. However, $L$ exhibits diminishing returns: **1$\rightarrow$3 stages gives −20.5%**, 3$\rightarrow$5 adds −11.8%, and 5$\rightarrow$10 contributes only −2.1% further—so the reported OmniField variant where we use **3 stages strikes the balance** between performance and compute.

---

> ### Author Response · Authors · 2025-11-25
>
> **6. Diversity of downstream tasks beyond reconstruction and forecasting**
>
> **Summary**: OmniField is not limited to reconstruction and forecasting; we have added experiments showing its utility for (1) classification, (2) anomaly detection, and (3) (non-spatial) time series. We will soon incorporate these results into the main text (with details in the appendix) to make clear that OmniField supports additional spatiotemporal tasks on top of the reconstruction and forecasting tasks currently emphasized.
>
>
> **(1) Classification**: We evaluate OmniField as a **representation learner** for classification under sparse observation. On CIFAR-10, we randomly subsample 20% of pixels per image and keep this pattern fixed across epochs. OmniField is first trained for reconstruction using 10% of pixels as conditional input and 10% as supervised targets. Using a separate classifier (ResNet-50), we then use the learned representation for the CIFAR-10 classification. Our findings are as follows:
> - Trained from scratch, OmniField and the classifier records 57.24% and 55% accuracy, respectively.
> - A foundation model approach, where the pretrained OmniField’s **feature** (*g* in Fig. 2b) is used as input for the classifier, achieves 62.5% accuracy, improving +13.6% upon training from scratch.
> - A fine-tuning approach, where the pretrained OmniField is fine-tuned using a classification head (i.e., projection head) achieves 71.83%, improving +30.6% against the baseline.
> These experiments highlight potentials of our proposed OmniField training as a novel pretext task, which merits future explorations.
>
> **(2) Anomaly detection (SEVIR lightning prediction)**: We further evaluate OmniField on SEVIR [1], a multimodal storm-event imagery dataset with four input modalities and sparse lightning-flash labels. **Settings**: We use the publicly available SEVIR dataset, for which all four input modalities are downsampled to 24×24 (lightning remains the sparse target), and we use the paper’s 60/20/20 train/val/test split. Models are end-to-end trained to predict lightning occurrence directly from the multimodal fields (denoted *g* in Fig. 2d), and we report ROC–AUC on the test set. **Training & results**: we train all models as binary classifiers. Under this common setup, the original UNet/CNN baseline and OmniField both reach AUC=0.97, while FNO achieves 0.96, showing that OmniField’s multimodal fusion and refinement extends robustly to anomaly detection.
>
> **(3)  Non-geospatial multimodal time series (CESNET)**: When spatial coordinates are removed and the task is purely temporal, **OmniField still outperforms prior arts on a time-series data**. Settings: We train both OmniField and FNO on the CESNET [2] multivariate/multimodal forecasting task using the official data pipeline and objective. Concretely, we aggregate raw events into 1-hour bins and retain the 10 numeric metrics as used by TIME-IMM [3], then form sliding windows of 7 days of context followed by forecasting 7 days of future targets with a 1 hour stride. Both models are then trained with a batch size of 64 to minimize MSE over all 10 output variables and all forecast steps. Results: Continuous neural field (CNF) models **achieve new state-of-the-art performance on CESNET**. The FNO baseline obtains a test MSE of 0.790, outperforming strong non-CNF multimodal forecasters such as TimeMixer[4], TimesNet[5], CRU[6], and Informer [7]. Our **OmniField** further improves the CNF frontier to an MSE of 0.755, outperforming all prior methods, corresponding to **12% lower error than the best existing prior art** (TimeMixer) and 4% lower error than the FNO baseline. A summary of the comparison is shown below:
>
> | Model | OmniField | FNO | TimeMixer | TimesNet | CRU | Informer |
> |--|--|--|--|--|--|--|
> | MSE|**0.755**|*0.790*|0.859|0.869|0.873|0.875|
>
> **We appreciate your thoughtful feedback and hope the clarifications provided show the value of our contributions and justify a higher score.**
>
> [1] M. Veillette et al. "Sevir: A storm event imagery dataset for deep learning applications in radar and satellite meteorology." NeurIPS 2020
>
> [2] J. Koumar et al. CESNET-TimeSeries24: Time Series Dataset for Network Traffic Anomaly Detection and Forecasting. Sci Data 12, 338 (2025)
>
> [3] C. Chang et al., Time-IMM: A Dataset and Benchmark for Irregular Multimodal Multivariate Time Series, NeurIPS 2025.
>
> [4] S. Wang et al., TimeMixer: Decomposable Multiscale Mixing for Time Series Forecasting, ICLR 2024.
>
> [5] H. Wu et al., TimesNet: Temporal 2D-Variation Modeling for General Time Series Analysis, ICLR 2023.
>
> [6] J. Schirmer et al., Modeling Irregular Time Series with Continuous Recurrent Units, ICML 2022.
>
> [7] H. Zhou et al., Informer: Beyond Efficient Transformer for Long Sequence Time-Series Forecasting, AAAI 2021.

---

### Official Review · Reviewer_JbX6 · 2025-10-31

**Soundness:** 3
**Presentation:** 4
**Contribution:** 3
**Rating:** 6
**Confidence:** 3

**Summary:**

This work presents a unified neural field paradigm that tackles the challenge of learning continuous, frequency-rich representations across heterogeneous spatial, temporal, and multimodal sensing regimes.
The authors blend gated fleximodal fusion with iterative cross-modal refinement, allowing the model to gracefully interpolate missing channels, suppress noisy sensors, and sustain high-frequency fidelity in sparse, real-world observation settings.
The authors evaluated the proposed method using climate simulation (ClimSim-THW), air-quality forecasting (EPA-AQS), and vision-centric reconstruction tasks, compared with baselines like SCENT and RainNet.

**Strengths:**

- The proposed fleximodal fusion and iterative refinement approach provides a principled mechanism for handling missing and noisy modalities, improving robustness in settings with irregular or sparse sensors.
- The incorporation of frequency-rich embeddings and sinusoidal initialization yields measurable gains in high-frequency signal reconstruction, particularly in spatiotemporal domains.
- The method showed consistent performance improvements across multiple scientific datasets, suggesting a generalizable modeling framework rather than task-specific tuning.

**Weaknesses:**

- The evaluation focuses on a curated set of scientific benchmarks; broader assessment on diverse multimodal domains (robotics, remote sensing beyond climate/air quality) would strengthen claims of generality.
- The computational and memory cost of iterative cross-modal refinement and continuous-field conditioning is not fully characterized. It's unclear how well the proposed method scales to higher-resolution or real-time applications.
- While robustness to missing modalities is a central motivation, ablation analyses on modality-specific contributions are limited.

**Questions:**

See weaknesses.

---

> ### Author Response · Authors · 2025-11-25
>
> We appreciate your constructive reviews.
>
> **1. Diversity of Multimodal Domains**
>
> > ... broader assessment on diverse multimodal domains (robotics, remote sensing beyond climate/air quality) would strengthen claims of generality.
>
> Remote sensing and robotics are both domains where we believe OmniField can be particularly impactful. To highlight OmniField’s broader applicability, we additionally evaluate it on diverse multimodal scenarios beyond reconstruction and forecasting, including (1) anomaly detection (on a **remote sensing** dataset), (2) non-spatial time-series, and (3) classification. We view robotics as a natural and important extension of this line of work, and we are indeed beginning to explore it as a future work.
>
> **(1) Anomaly detection (SEVIR lightning prediction)**: We evaluate OmniField on a remote-sensing scenario using SEVIR [1], a multimodal storm-event imagery dataset with four input modalities and sparse lightning-flash labels. **Settings**: We use the publicly available SEVIR dataset, for which all four input modalities are downsampled to 24×24 (lightning remains the sparse target), and we use the paper’s 60/20/20 train/val/test split. Models are end-to-end trained to predict lightning occurrence directly from the multimodal fields (denoted *g* in Fig. 2d), and we report ROC–AUC on the test set. **Training & results**: we train all models as binary classifiers. Under this common setup, the original UNet/CNN baseline and OmniField both reach AUC=0.97, while FNO achieves 0.96, showing that OmniField’s multimodal fusion and refinement extends robustly to anomaly detection.
>
> **(2)  Non-geospatial multimodal time series (CESNET)**: When spatial coordinates are removed and the task is purely temporal, **OmniField still outperforms prior arts on a time-series data**. **Settings**: We train both OmniField and FNO on the CESNET [2] multivariate/multimodal forecasting task using the official data pipeline and objective. Concretely, we aggregate raw events into 1-hour bins and retain the 10 numeric metrics as used by TIME-IMM [3], then form sliding windows of 7 days of context followed by forecasting 7 days of future targets with a 1 hour stride. Both models are then trained with a batch size of 64 to minimize MSE over all 10 output variables and all forecast steps. **Results**: Continuous neural field (CNF) models **achieve new state-of-the-art performance on CESNET**. The FNO baseline obtains a test MSE of 0.790, outperforming strong non-CNF multimodal forecasters such as TimeMixer[4], TimesNet[5], CRU[6], and Informer [7]. Our **OmniField** further improves the CNF frontier to an MSE of 0.755, outperforming all prior methods, corresponding to **12% lower error than the best existing prior art** (TimeMixer) and 4% lower error than the FNO baseline. A summary of the comparison is shown below:
>
> | Model | OmniField | FNO | TimeMixer | TimesNet | CRU | Informer |
> |--|--|--|--|--|--|--|
> | MSE|**0.755**|*0.790*|0.859|0.869|0.873|0.875|
>
> **(3) Classification**: We evaluate OmniField as a **representation learner** for classification under sparse observation. On CIFAR-10, we randomly subsample 20% of pixels per image and keep this pattern fixed across epochs. OmniField is first trained for reconstruction using 10% of pixels as conditional input and 10% as supervised targets. Using a separate classifier (ResNet-50), we then use the learned representation for the CIFAR-10 classification. Our findings are as follows:
> - **Trained from scratch**, OmniField and the classifier records 57.24% and 55% accuracy, respectively.
> - **A foundation model approach**, where the pretrained OmniField’s **feature** (*g* in Fig. 2b) is used as input for the classifier, achieves 62.5% accuracy, improving +13.6% upon training from scratch.
> - **A fine-tuning approach**, where the pretrained OmniField is fine-tuned using a classification head (i.e., projection head) achieves 71.83%, improving +30.6% against the baseline.
> These experiments highlight potentials of our proposed OmniField training as a novel pretext task, which merits future explorations.
>
> [1] M. Veillette et al. SEVIR: A Storm Event Imagery Dataset for Deep Learning Applications in Radar and Satellite Meteorology. NeurIPS 2020
>
> [2] J. Koumar et al. CESNET-TimeSeries24: Time Series Dataset for Network Traffic Anomaly Detection and Forecasting. Sci Data 12, 338 (2025)
>
> [3] C. Chang et al., Time-IMM: A Dataset and Benchmark for Irregular Multimodal Multivariate Time Series, NeurIPS 2025.
>
> [4] S. Wang et al., TimeMixer: Decomposable Multiscale Mixing for Time Series Forecasting, ICLR 2024.
>
> [5] H. Wu et al., TimesNet: Temporal 2D-Variation Modeling for General Time Series Analysis, ICLR 2023.
>
> [6] J. Schirmer et al., Modeling Irregular Time Series with Continuous Recurrent Units, ICML 2022.
>
> [7] H. Zhou et al., Informer: Beyond Efficient Transformer for Long Sequence Time-Series Forecasting, AAAI 2021.

---

> ### Author Response · Authors · 2025-11-25
>
> > The computational and memory cost of iterative cross-modal refinement and continuous-field conditioning is not fully characterized.
>
> **We agree** that computational and memory cost are crucial aspects to consider. Our additional experiments highlight three key points:
>
>
> - **Compute Efficiency**: OmniField is markedly more accurate and consistently achieves a better compute (measured in FLOPs)–performance tradeoff than baseline models.
>
>
> - **Memory Efficiency**: The smallest OmniField variant (1.1M parameters) records substantially higher performance than the largest FNO or PROSE-FD variants (38.1M and 38.3M parameters, respectively).
>
>
> - **Cost of ICMR**: Ablations and a time-complexity analysis show that increasing the number of refinement iterations incurs only a modest increase in compute (FLOPs). Additional iterations monotonically improve performance.
>
>
>
> **2. On Multimodal Efficiency for Continuous-Field Conditioning**
>
>
> **Table 1** – Efficiency and Performance Comparisons
> |**Model**|**Input mods**|**Params** (M)|**FLOPs** (fwd)|**T** (err)|**H** (err)|**W** (err)|
> |---|---|---|---|---|---|---|
> |FNO|T|10.6|5.0e9|**1.10**|*0.98*|6.48|
> |PROSE-FD|T|10.5|8.8e11|1.28|1.30|*5.76*|
> |OmniField|T|10.6|2.8e10|*1.11*|**0.76**|**5.20**|
> | | | | | | | |
> |FNO|THW|1.1|1.6e9|3.36|1.39|7.19|
> |FNO|THW|10.6|5.0e9|2.96|1.16|6.79|
> |FNO|THW|38.1|1.7e11|2.53|1.13|6.72|
> |PROSE-FD|THW|1.0|3.2e11|5.79|2.31|6.32|
> |PROSE-FD|THW|10.5|3.1e12|5.40|1.95|5.66|
> |PROSE-FD|THW|38.3|6.4e12|3.03|1.18|5.71|
> |OmniField|THW|1.3|1.6e10|1.21|0.74|5.13|
> |OmniField|THW|10.6|3.0e10|*1.10*|*0.68*|*4.94*|
> |OmniField|THW|37.4|3.6e10|**1.07**|**0.66**|**4.86**|
>
>
> We compare OmniField to FNO and PROSE-FD at matched parameter counts for **unimodal** (T) and **multimodal** (THW) settings.
>
>
>
> **Unimodal (T)**: All models have \~10M parameters, but FNO is cheapest (5.0e9 FLOPs), OmniField is moderately heavier (2.8e10 FLOPs), and PROSE-FD is two orders of magnitude more expensive (8.8e11 FLOPs). OmniField nonetheless attains the best or comparable accuracy, with lower H and W error than both baselines and T error on par with FNO, showing a favorable accuracy–efficiency trade-off against both baselines.
>
>
>
> **Multimodal (THW)**: At 1M parameters, OmniField already outperforms FNO and PROSE-FD by a wide margin on all variables (e.g., T error 1.21 vs. 3.36 for FNO and 5.79 for PROSE-FD), using ~10× the FLOPs of FNO but >20× fewer FLOPs than PROSE-FD. As capacity scales to 10M and 38M parameters, OmniField continues to improve, reaching T/H/W errors of 1.07/0.66/4.86 at 37.4M parameters with 3.6e10 FLOPs—over an order of magnitude cheaper than 38M PROSE-FD (6.4e12 FLOPs) and cheaper than the largest FNO (1.7e11 FLOPs), while maintaining significantly better accuracy. Unlike FNO and PROSE-FD, where adding modalities can **degrade** performance, OmniField consistently exploits H and W to reduce error without a proportional compute increase. Overall, at matched capacity, OmniField offers a superior efficiency–accuracy trade-off, particularly in the multimodal regime.
>
>
> **3. On ICMR Refinements**
>
>
>
> **Time Complexity Comparisons.** Let $M$ be the number of modalities (e.g., $1 \rightarrow 3 \rightarrow 6$), $D$ the latent/feature width (FNO: channel width $C$; PROSE-FD/OmniField: attention width $d$), and $L$ the number of ICMR refinement iterations (OmniField only).
>
>
>
> | **Model** | **Big-O** | **Intuition** |
> |---|---|---|
> | FNO | $O(MD)$ | Modalities contribute linearly to channel count; FFT and linear projections scale with width. |
> | PROSE-FD |$O(M^{2}D$) | Single-pass self-attention over fused multimodal tokens. |
> | OmniField | $O(LM^{2}D)$ | Each ICMR stage revisits multimodal fusion; pairwise cross-modal interactions lead to the $M^{2}$ term. |
>
>
> Theoretically, OmniField’s cost grows linearly with $L$. In practice, FLOPs grow *sublinearly* with $L$ because many components (projections, decoders) do not scale with refinement depth. This is evident in the following empirical result.
>
>
> |**ICMR stages**|**Params** (M)|**FLOPs** (forward)|**T** (err)|**H** (err)|**W** (err)|
> |---|---|---|---|---|---|
> |1|5.4|2.82e+10|1.383|0.905|5.289|
> |3|10.6|3.0e+10|1.10|0.68|4.94|
> |5|18.5|3.2e+10|*0.97*|*0.61*|*4.86*|
> |10|34.4|3.8e+10|**0.95**|**0.60**|**4.83**|
>
>
> Scaling $L$ from 1$\rightarrow$10 stages (5.4M$\rightarrow$34.4M params) improves T RMSE 1.383$\rightarrow$0.95 (−31%) with only +35% FLOPs (2.82e10$\rightarrow$3.8e10), whereas widening the model width (Table 1; 10.6M$\rightarrow$37.4M params) yields just 1.10$\rightarrow$1.07 (−2.7%) for +20% FLOPs—highlighting that $L$ is the more effective lever at comparable parameter budgets and supports the pivotal role of ICMR. However, $L$ exhibits diminishing returns: **1$\rightarrow$3 stages gives −20.5%**, 3$\rightarrow$5 adds −11.8%, and 5$\rightarrow$10 contributes only −2.1% further—so the reported OmniField variant where we use **3 stages strikes the balance** between performance and compute.

---

> ### Author Response · Authors · 2025-11-25
>
> **4. Response to scalability concern.**
> > It's unclear how well the proposed method scales to higher-resolution or real-time applications
>
> We agree that scalability to very high-resolution or real-time settings is an important question, and we clarify that **OmniField is explicitly designed to avoid quadratic dependence on the number of input tokens**. Our backbone is **Perceiver IO** [1], where a single pixel (or sensor reading) is treated as a single token. In our design, the only component whose cost scales with the number of input tokens $N$ is the unimodal encoder: a fixed-size latent set (of size $K$) attends over all tokens, giving a complexity of $\mathcal{O}(N \cdot K \cdot d)$, which is effectively **linear in $N$ since $K$ is constant**. In contrast, both MCT and ICMR operate purely in the fixed-size latent space, so their cost is independent of $N$. As a result, the overall computational complexity of **OmniField scales linearly with the number of input tokens $N$**, and no $N^2$ term is introduced by MCT or ICMR.
>
>
> Compared to SCENT, which also inherits Perceiver IO but retains self-attention structures in both encoder and decoder (leading to higher effective cost when operating directly in token space), **OmniField keeps all cross-modal fusion and refinement in the latent domain**, making it more scalable with respect to the number of observed tokens. Moreover, our target regime is sparse scientific sensing, where the number of observed tokens is much smaller than the full spatiotemporal grid, further reducing the effective computational burden. We will clarify these complexity properties in the main text and explicitly note that fully dense, extremely high-resolution scientific fields remain a limitation of the current architecture and an important direction for future work (e.g., via hierarchical or patch-based tokenization for real-time deployment).
>
>
> **5. Further Notes on Multimodal Contributions**
>
> > While robustness to missing modalities is a central motivation, ablation analyses on modality-specific contributions are limited.
>
> We emphasize modality-specific contributions as follows:
> - **More modalities always help for OmniField**: We studied modality-specific contributions in Fig. 4 (ClimSim T vs THW, EPA-AQS 2/4/6 pollutants) and report that adding modalities consistently reduces per-target error. However, we will point this out more explicitly and reference the corresponding tables.
>
>
> - **Gains from additional modalities are not always obvious for the baselines**. Additional experiments in Table 1 shows that baseline performances can degrade due to multimodality. All baselines are not made for realistic challenges of **multimodal** sparse and irregularly sampled observations, and we conjecture that the multimodal correspondence structure from our real-world datasets would be much more complex than controlled / simulated environments. Multimodal inputs are not always correlated and can be noisy – our MCT and ICMR architectures implicitly enables this selective aggregation of useful information while discarding noise. This finding is consistent in Fig. 4b on EPA-AQS dataset, where additional modalities often harm performance.
>
>
> - **Randomly missing modalities during training barely harms performance for OmniField.** To test robustness to randomly missing modalities during training, we train the 37M-parameter model with three ICMR blocks while, for 100·p% of training instances, randomly dropping one or two modalities (but never all three). As shown in Table 2, performance is almost unchanged when p = 0.5, and even at p = 0.75 the degradation is modest (T, H, and W increase by only 0.16, 0.08, and 0.08, respectively, relative to p = 0), indicating that the fusion mechanism handles missing inputs without leaking information from absent modalities. Finally, note that EPA-AQS naturally has missing modalities both in training and testing, serving as a realistic testbed of missing modalities.
>
>
> **Table 2: Performance under randomly missing modalities**
>
>
> | **p** | **T** | **Q**  | **V** |
> |---|---|---|---|
> | 0.00 | 1.07 | 0.66 | 4.86 |
> | 0.50 | 1.11 | 0.71 | 4.89 |
> | 0.75 | 1.23 | 0.74 | 4.94 |
>
> [1] A. Jaegle et al. Perceiver IO: A General Architecture for Structured Inputs & Outputs. arXiv preprint arXiv:2107.14795.

---

### Official Review · Reviewer_BeG4 · 2025-10-31

**Soundness:** 4
**Presentation:** 3
**Contribution:** 2
**Rating:** 6
**Confidence:** 2

**Summary:**

The paper proposes OmniField, a framework for robust multimodal spatiotemporal learning from sparse, irregular, and noisy real-world observational data.
OmniField extends the continuous neural field paradigm (previously introduced in SCENT) by addressing two fundamental challenges: data sparsity and modality inconsistency.
Specifically, it integrates several components — Gaussian Fourier Features (GFF) and Sinusoidal Initialization (SI) for improved spatial continuity, and three new modules:
MCT (Multimodal Crosstalk) for early cross-modal feature interaction,
ICMR (Iterative Cross-Modal Refinement) for multi-round alignment across modalities,
and Fleximodal Fusion for handling missing or degraded modalities.
Extensive experiments on four datasets (CIFAR-10, RainNet, ClimSim-THW, and EPA-AQS) and eight baselines show consistent performance improvements in both reconstruction and forecasting tasks, especially under conditions of high sparsity and missing modalities.

**Strengths:**

* The paper is well-structured and clearly motivates the problem by identifying two central challenges — data sparsity and multimodal inconsistency — and proposing targeted solutions through the MCT, ICMR, and Fleximodal Fusion modules. The organization is logical.

* The figures are well-designed and self-explanatory, effectively supporting the paper’s claims and illustrating the benefits of the proposed modules. Quantitative results are straightforward and convincing, showing consistent gains over baselines.

* The experimental results are solid and comprehensive, covering eight strong baselines and four diverse datasets. The results are straightforward to interpret and show consistent improvements across all settings.

* The paper includes extensive ablation studies, noise-robustness analyses, and missing-modality evaluations, which together demonstrate strong empirical evidence for the proposed method's robustness and generality.

**Weaknesses:**

* While the paper is generally well written, some parts are conceptually dense and abstract. The presentation could benefit from additional intuition, clearer intermediate explanations, or a small running example to illustrate how each proposed component (MCT, ICMR, Fleximodal Fusion) functions in practice.
* The forecasting horizon studied in the current experiments is relatively short (e.g., six-hour prediction on ClimSim-THW). Evaluating longer temporal horizons could provide deeper insights into the model’s stability and long-term reasoning ability. Incorporating additional temporal modeling structures might further enhance the method’s forecasting capability.
* OmniField introduces additional modules (MCT and three rounds of ICMR), which substantially increase the computational cost compared to the baselines. While the model achieves lower RMSE and improved robustness, these gains come with notably higher architectural complexity. A more detailed analysis of runtime efficiency trade-offs would help clarify the impact of the proposed design.

**Questions:**

* In Figure 4(b–c), the label “SCENT-MM” seems to correspond to OmniField, based on the text and tables. Could you please confirm whether this is a labeling oversight, or if SCENT-MM refers to a distinct model configuration?
* I noticed that OmniField has a relatively large number of parameters compared to SCENT. To what extent might the performance improvement stem from model capacity rather than the proposed architectural changes?

---

> ### Author Response · Authors · 2025-11-26
>
> **1. Improving the Presentation**
> >The presentation could benefit from additional intuition, clearer intermediate explanations, or a small running example to illustrate how each proposed component (MCT, ICMR, Fleximodal Fusion) functions in practice.
>
>
>
>
> We thank the reviewer for this suggestion. In the camera-ready version, we will add (i) concise intuitive summaries for MCT, ICMR, and Fleximodal Fusion, (ii) a compact running air-quality example illustrating how each component operates on partially overlapping $O_{3}$, $PM_{2.5}$, and $NO_{2}$ stations, and (iii) clearer intermediate explanations linking the formal definitions (e.g., Eq. 1) to the model’s behavior in practice.
>
>
>
>
> **A Running Example.**  Consider three pollutants, $O_{3}$, $PM_{2.5}$, and $NO_{2}$ measured by partially overlapping station networks across the U.S. On a given day, some $O_{3}$ stations may be offline, some $PM_{2.5}$ measurements may be removed by QA/QC, and a subset of locations may report all three pollutants. The task is to use all available measurements to forecast the full $O_{3}$, $PM_{2.5}$, and $NO_{2}$ fields 1~6 days ahead, including at locations with no direct observations. OmniField encodes each pollutant’s stations into modality-specific tokens, enables cross-modal interaction via multimodal crosstalk (MCT), iteratively refines a shared global representation with ICMR, and decodes pollutant-specific forecasts. Fleximodal fusion handles missing modalities so that a single model can operate on any subset of sensors.
>
>
>
>
> **MCT Intuition.**  In this setting, MCT enables $O_{3}$, $PM_{2.5}$, and $NO_{2}$ tokens to exchange information prior to decoding. Each modality first encodes its own station observations into a set of tokens. MCT then concatenates tokens from all present modalities and conditions them on a global feature vector $z$ that summarizes the joint state. This operation allows, for example, $O_{3}$ tokens to attend to nearby $PM_{2.5}$ and $NO_{2}$ tokens, leveraging co-located and spatially proximate measurements and improving alignment between modalities that live on different supports and scales.
>
>
>
>
> **ICMR Intuition.**  A single MCT pass may be insufficient to reconcile noisy or partially inconsistent modalities. ICMR treats $z$ as a global latent summary that is updated iteratively. At each MCT block, modalities exchange information via attention to produce fused token representations $h^{(k)}$, after which $z^{(k+1)}$ is updated (e.g., by averaging these tokens). Over multiple iterations, this global summary becomes more coherent and informative, enabling subsequent MCT passes to place greater weight on reliable sensors and reduce the influence of corrupted or outlying measurements. In the air-quality setting, this allows the model to rely more heavily on cleaner $O_{3}$ and $NO_{2}$ observations when $PM_{2.5}$ is degraded.
>
>
>
>
> **Fleximodal Fusion Intuition.**  Fleximodal fusion allows OmniField to be applied with any subset of modalities without retraining. A binary presence mask $\pi_m$ indicates which modalities are available for a given instance. If $PM_{2.5}$ is absent on a particular day, its encoder outputs and associated cross-modal interactions are gated off, and $PM_{2.5}$ is excluded from the loss for that instance. This design ensures that the model does not rely on unobserved channels and that a single set of parameters can be used for fully observed, partially observed, and single-modality inputs.

---

> ### Author Response · Authors · 2025-11-26
>
> > Evaluating longer temporal horizons could provide deeper insights into the model’s stability and long-term reasoning ability. Incorporating additional temporal modeling structures might further enhance the method’s forecasting capability.
>
>
> **2. Longer Time Horizon Prediction**
>
>
> We thank the reviewer for this constructive comment. Our **OmniField architecture already incorporates an explicit temporal modeling structure designed for long-horizon forecasting**: it inherits the *warp-unrolling forecasting (WUF)* strategy from SCENT. WUF is built on a continuous-time latent *Time Warp Processor* that is **trained to advance the latent state by any time increment $\Delta t$ within a maximum time horizon $t_h$**. At inference, instead of naively unrolling one step at a time (which rapidly accumulates errors), we *warp* directly to a long time horizon $t_h$ and then perform only a small number of additional prediction steps from this warped reference state. This ensures that each forecasted frame is at most a few prediction steps away from a supervised state, substantially reducing error accumulation and stabilizing long-range rollouts.
>
>
> To directly address the reviewer’s request, we trained OmniField on ClimSim-THW,  both on a short $6$-hour horizon and a long $48$-hour horizon, and evaluated all models at $6$, $24$, and $48$ hours. We also compared against an FNO baseline unrolled for the same number of steps. The table below reports errors for three variables ($T$ / $H$ / $W$; lower is better). FNO errors are shown in scientific notation (effectively requiring a log scale to visualize) because its rollout diverges at longer horizons.
>
>
> | Model       | Train horizon | Eval horizon | $T$                     | $H$                     | $W$                      |
> |------------|---------------|--------------|-------------------------|-------------------------|--------------------------|
> | OmniField  | $48$ h        | $6$ h        | **$0.056$**              | **$0.084$**              | **$0.700$**               |
> | OmniField  | $48$ h        | $24$ h       | $0.056$              | $0.092$              | $0.722$               |
> | OmniField  | $48$ h        | $48$ h       | $0.063$              | $0.095$              | $0.754$               |
> | OmniField  | $6$ h         | $6$ h        | *$0.062$*              | *$0.093$*              | *$0.707$*               |
> | OmniField  | $6$ h         | $24$ h       | $0.085$              | $0.134$              | $0.728$               |
> | OmniField  | $6$ h         | $48$ h       | $0.087$              | $0.138$              | $0.789$               |
> | FNO        | --            | $6$ h        | $0.449$   | $0.417$   | $1.860$     |
> | FNO        | --            | $24$ h       | $2.58 \times 10^{3}$    | $2.66 \times 10^{3}$    | $1.66 \times 10^{4}$     |
> | FNO        | --            | $48$ h       | $9.36 \times 10^{9}$    | $1.01 \times 10^{10}$   | $6.60 \times 10^{10}$    |
>
>
> These results demonstrate that:
>
>
> - **OmniField trained on a $48$-hour horizon strictly outperforms the model trained on a $6$-hour horizon at all evaluation horizons.** For example, at $48$ hours, $T$-error improves from $0.087$ (trained on $6$ h) to $0.063$ (trained on $48$ h), and similar gains hold for $H$ and $W$.
>
>
> - **The $48$-hour OmniField model’s performance degrades very gracefully as the forecast horizon increases.** For $T$, the error only changes from $0.056 \rightarrow 0.056 \rightarrow 0.063$ as we move from $6 \rightarrow 24 \rightarrow 48$ hours, showing a smooth, stable dependence on horizon length rather than catastrophic error accumulation.
>
>
> - **FNO unrolling for $48$ steps blows up.** Even at $24$ hours, FNO’s errors already reach $\mathcal{O}(10^{3})$, and by $48$ hours they explode to $\mathcal{O}(10^{10})$, making meaningful long-horizon forecasting impossible. This divergence is precisely what WUF is designed to avoid: by warping in time and limiting the effective number of unrolled steps, OmniField maintains stable and accurate long-horizon forecasts, directly addressing the reviewer’s concern about stability and long-term reasoning.

---

> ### Author Response · Authors · 2025-11-26
>
> > … A more detailed analysis of runtime efficiency trade-offs would help clarify the impact of the proposed design.
>
>
>
>
> We appreciate the reviewer for this suggestion. While our initial baseline model sizes reflect respective original implementations, we strongly agree that computational and memory cost are crucial aspects to consider. Our additional experiments highlight three key points:
>
>
>
>
> - **Runtime Efficiency**: OmniField is markedly more accurate and consistently achieves a better runtime (measured in FLOPs)–performance tradeoff than baseline models.
>
>
>
>
> - **Memory Efficiency**: The smallest OmniField variant (1.1M parameters) records substantially higher performance than the largest FNO, PROSE-FD, or SCENT variants (38.1M, 38.3M, and 29.3M parameters, respectively).
>
>
>
>
> - **Cost of ICMR**: Ablations and a time-complexity analysis show that increasing the number of refinement iterations incurs only a modest increase in compute (FLOPs). Additional iterations monotonically improve performance, but with diminishing returns, clarifying how to choose an efficient operating point.
>
>
>
>
> **3. Efficiency Trade-offs**
>
>
>
>
> **Table 1** – Efficiency and Performance for Multimodal Setting
> |**Model**|**Input mods**|**Params** (M)|**FLOPs** (fwd)|**T** (err)|**H** (err)|**W** (err)|
> |---|---|---|---|---|---|---|
> |FNO|THW|1.1|1.6e9|3.36|1.39|7.19|
> |FNO|THW|10.6|5.0e9|2.96|1.16|6.79|
> |FNO|THW|38.1|1.7e11|2.53|1.13|6.72|
> |PROSE-FD|THW|1.0|3.2e11|5.79|2.31|6.32|
> |PROSE-FD|THW|10.5|3.1e12|5.40|1.95|5.66|
> |PROSE-FD|THW|38.3|6.4e12|3.03|1.18|5.71|
> |**OmniField**|THW|1.3|1.6e10|1.21|0.74|5.13|
> |**OmniField**|THW|10.6|3.0e10|*1.10*|*0.68*|*4.94*|
> |**OmniField**|THW|37.4|3.6e10|**1.07**|**0.66**|**4.86**|
> |**SCENT**|THW|29.3|4.4e10|1.52|0.99|5.07|
>
>
>
>
> **Efficiency**: At 1M parameters, OmniField already outperforms the largest FNO, PROSE-FD, and SCENT variants by a wide margin on all T, H, W variables. Unlike FNO and PROSE-FD, where adding modalities can **degrade** performance, OmniField consistently exploits H and W to reduce error without a proportional compute increase. Overall, at matched capacity, OmniField offers a superior efficiency–accuracy trade-off, particularly in the multimodal regime.
>
>
>
>
>
>
>
>
> **4. On the Number of ICMR Refinements**
>
>
>
>
> **Time Complexity Comparisons.** Let $M$ be the number of modalities (e.g., $1 \rightarrow 3 \rightarrow 6$), $D$ the latent/feature width (FNO: channel width $C$; PROSE-FD/OmniField: attention width $d$), and $L$ the number of ICMR refinement iterations (OmniField only).
>
>
>
>
> | **Model** | **Big-O** | **Intuition** |
> |---|---|---|
> | FNO | $O(MD)$ | Modalities contribute linearly to channel count; FFT and linear projections scale with width. |
> | PROSE-FD |$O(M^{2}D$) | Single-pass self-attention over fused multimodal tokens. |
> | OmniField | $O(LM^{2}D)$ | Each ICMR stage revisits multimodal fusion; pairwise cross-modal interactions lead to the $M^{2}$ term. |
>
>
>
>
> Theoretically, OmniField’s cost grows linearly with $L$. In practice, FLOPs grow *sublinearly* with $L$ because many components (projections, decoders) do not scale with refinement depth.
>
>
>
>
> |**ICMR stages**|**Params** (M)|**FLOPs** (forward)|**T** (err)|**H** (err)|**W** (err)|
> |---|---|---|---|---|---|
> |1|5.4|2.82e+10|1.383|0.905|5.289|
> |3|10.6|3.0e+10|1.10|0.68|4.94|
> |5|18.5|3.2e+10|*0.97*|*0.61*|*4.86*|
> |10|34.4|3.8e+10|**0.95**|**0.60**|**4.83**|
>
>
>
>
> Scaling $L$ from 1$\rightarrow$10 stages (5.4M$\rightarrow$34.4M params) improves T RMSE 1.383$\rightarrow$0.95 (−31%) with only +35% FLOPs (2.82e10$\rightarrow$3.8e10), whereas widening the model width (Table 1; 10.6M$\rightarrow$37.4M params) yields just 1.10$\rightarrow$1.07 (−2.7%) for +20% FLOPs—highlighting that $L$ is the more effective lever at comparable parameter budgets and supports the pivotal role of ICMR. However, $L$ exhibits diminishing returns: **1$\rightarrow$3 stages gives −20.5%**, 3$\rightarrow$5 adds −11.8%, and 5$\rightarrow$10 contributes only −2.1% further—so the reported OmniField variant where we use **3 stages strikes the balance** between performance and compute.
>
>
>
>
>
>
>
>
> > I noticed that OmniField has a relatively large number of parameters compared to SCENT. To what extent might the performance improvement stem from model capacity rather than the proposed architectural changes?
>
>
>
>
> **Against SCENT**: Table 1 suggests that OmniField with 1.3M parameters outperforms SCENT with 29.3M parameters. In the meanwhile, SCENT with 29.3M parameters has larger FLOPs than OmniField with 37.4M parameters due to the expensive sparse attention layers of SCENT (4.4e10 vs. 3.6e10). Also, the ICMR refinement iterations ($L$) largely determines the performance. Judging by these results, we conclude that performance improvement stems from the proposed architectural changes including MCT, ICMR and the ICMR depth ($L$), rather than model capacity.

---

### Author Response · Authors · 2025-11-25

***Dear Reviewers***

Thank you for the thoughtful and detailed reviews. Your comments prompted us to strengthen our paper including efficiency analysis and additional ablations, and broaden the paper’s scope with additional tasks and clearer presentation. Here **we provide a brief summary of our new experiments and findings**. Notably, we showcase the following results:

**(1) Efficiency under multimodal training**. Firstly, with comparable parameter budgets, **OmniField consistently occupies a better compute–performance point than baselines**: it is markedly more accurate than FNO while remaining far cheaper than PROSE-FD. Second, our ablation shows that increasing ICMR iterations (*L*) is the most effective lever: moving from 1→10 stages improves RMSE (for *T*) 1.383→0.95 (−31%) for only +35% FLOPs, whereas increasing the model width (10.6M→37.4M params) yields only 1.10→1.07 (−2.7%) at +20% FLOPs. We also find that increasing *L* consistently improves performance, justifying the novelty in architecture design.

**(2) Task generalizability**. We added three multimodal domains to demonstrate that OmniField’s benefits transfer beyond reconstruction/forecasting as follows:

 • **Time-series forecasting** (CESNET [1]): OmniField achieves MSE 0.755, improving 13.8% upon strong time-series baselines (prior art: 0.859) and establishing state-of-the-art performance among the reported methods in a multimodal time series analysis.

 • **Classification under sparsity** (CIFAR-10, given 20% pixels): A UNet trained directly on sparse inputs attains 55.0% accuracy; a classifier on frozen OmniField features achieves 62.5%; a fine-tuned OmniField reaches 71.8%—evidence that OmniField’s learned continuous fields provide useful representations beyond the core tasks.

• **Anomaly detection** (SEVIR [2] lightning): We evaluate OmniField on a task to **predict lightning event from multimodal remote sensing**. Using a common preprocessing/evaluation pipeline, both the established UNet baseline and fine-tuned OmniField reaches 0.97 AUC. This shows that adapting OmniField to end-to-end anomaly detection is feasible and competitive.

**Overall**, these additional experiments underscore that OmniField is a versatile, generally applicable framework: it can be adapted to diverse modalities and objectives, supports multiple downstream tasks with a single architecture, and remains efficient across a wide range of operating points. **We again thank the reviewers for their insightful feedback, which helped us clarify and sharpen our claims**.

 **A Major Clarification**: We apologize for the naming inconsistency in Figure 4: “SCENT-MM” is an outdated name for the method we now call OmniField. The figure reports results for the same model, and we will update the label to “OmniField” in the final version.

[1] J. Koumar et al. CESNET-TimeSeries24: Time Series Dataset for Network Traffic Anomaly Detection and Forecasting. Sci Data 12, 338 (2025)

[2] M. Veillette et al. SEVIR: A Storm Event Imagery Dataset for Deep Learning Applications in Radar and Satellite Meteorology. NeurIPS 2020

---

### Meta-Review · Area_Chair_fLF7 · 2026-01-07

**Summary:**

Reviewers acknowledge strong empirical performance on reconstruction and forecasting tasks but raise concerns regarding:
- Limited conceptual novelty beyond prior continuous neural field frameworks (e.g., SCENT)
- Scalability---increasing architectural complexity introduced by multiple fusion and refinement modules
- Lack of broad applicability on downstream tasks or longer time horizon

After the rebuttal, two concerns remain somewhat less resolved. (1) The conceptual novelty beyond SCENT is not very well-pronounced, and the paper does not clearly articulate the unifying principle that elevates OmniField beyond a well-engineered aggregation of known components. (2) Downstream evaluations lack a strong baseline.

However, most concerns have been resolved well, and the merits seem to outweigh the flaws. I lean toward acceptance.

**Reviewer Concerns:**

- **Limited conceptual novelty.** Only partially resolved.
- **Scalability.** Successfully resolved via much efficiency analyses.
- **Lack of broad applicability.** Longer-horizon forecasting has been resolved, but not regarding the downstream tasks.

**Reviewer Scores:**

The other two reviewers might stay on the positive side, but the negative reviewer---BsBj---might have stayed on the negative side.

---

### Decision · Program_Chairs · 2026-01-26

Accept (Poster)